# A reinforcement learning approach to improve the performance of the Avellaneda-Stoikov market-making algorithm

**Javier Falces Marin** *, **David Díaz Pardo de Vera, Eduardo Lopez Gonzalo**

Escuela Técnica Superior de Ingenieros de Telecomunicación, SSR, Universidad Politécnica de Madrid, Madrid, Spain

* javifalces@gmail.com

**Data Availability Statement:** https://github.com/javifalces/HFTFramework.

**Funding:** The author(s) received no specific funding for this work.

## Abstract

Market making is a high-frequency trading problem for which solutions based on reinforcement learning (RL) are being explored increasingly. This paper presents an approach to market making using deep reinforcement learning, with the novelty that, rather than to set the bid and ask prices directly, the neural network output is used to tweak the risk aversion parameter and the output of the Avellaneda-Stoikov procedure to obtain bid and ask prices that minimise inventory risk. Two further contributions are, first, that the initial parameters for the Avellaneda-Stoikov equations are optimised with a genetic algorithm, which parameters are also used to create a baseline Avellaneda-Stoikov agent (Gen-AS); and second, that state-defining features forming the RL agent's neural network input are selected based on their relative importance by means of a random forest. Two variants of the deep RL model (Alpha-AS-1 and Alpha-AS-2) were backtested on real data (L2 tick data from 30 days of bitcoin–dollar pair trading) alongside the Gen-AS model and two other baselines. The performance of the five models was recorded through four indicators (the Sharpe, Sortino and P&L-to-MAP ratios, and the maximum drawdown). Gen-AS outperformed the two other baseline models on all indicators, and in turn the two Alpha-AS models substantially outperformed Gen-AS on Sharpe, Sortino and P&L-to-MAP. Localised excessive risk-taking by the Alpha-AS models, as reflected in a few heavy dropdowns, is a source of concern for which possible solutions are discussed.

## 1 Introduction

In securities markets, liquidity, that is, both the availability of assets for buyers and a demand for the same for sellers, is provided by market makers. (Foucault et al. [1] define liquidity more precisely as '*the degree to which an order can be executed within a short time frame at a price close to the consensus value of the security.' Conversely, a price that deviates substantially from this consensus value indicates illiquidity*.') Market makers provide liquidity as they exploit the market microstructure of orderbooks–which contain the minutest representation of trading data–where pending trade orders in a venue are placed in two price-ordered lists: a bid list

**Competing interests:** The authors have declared that no competing interests exist.

**Abbreviations:** T, Daily closing time; $t_j$, Current time instance (at arrival of the latest, the $j^{th}$, market tick); $\tau_i$, Time instance at the start of the $i^{th}$ 5-second action cycle of the RL agent; $p^m(t_j)$, Current market midprice (at time $t$); $I(t_j)$, Inventory held by the agent (at time $t_j$); $\gamma$, Risk aversion of the agent; $\sigma^2$, Variance of the market midprice; w, Size of window (in number of ticks) to estimate the variance of the market midprice; r, Reservation price; $\pi_n$, $n^{th}$ time interval for orderbook update rate calculation; $\delta^a$, $\delta^b$, Distance to the midprice from the reservation price on the ask ($\delta^a$) or bid ($\delta^b$) side; kna,knb, Liquidity parameter for the ask (kna) or bid (knb) side (for the $n^{th}$ time interval; $\lambda$na,$\lambda$nb, Arrival rate of orderbook updates on the ask ($\lambda$na) or bid ($\lambda$nb) side, for time interval $\pi_n$; $p^a$, $p^b$, Ask ($p^a$) or bid ($p^b$) price to be quoted; S, State space of the RL agent; A, Action space of the RL agent; R, Reward value of the RL algorithm; $\gamma_d$, Discount factor of the RL algorithm; α, Learning rate of the RL algorithm; s, Current state of the agent; s′, Prospective next state of the agent; a, Action taken by the agent from its current state; a′, Prospective next action of the agent; $Q_i(s, a)$, Q-value for state s and action a (at time $\tau_i$); $R(\tau_i)$, Asymmetric dampened P&L (at time $\tau_i$); $\Psi(\tau_i)$, Open P&L at time $\tau_i$; $\Delta m(\tau_i)$, Speculative P&L (the value difference between the open P&L and the close P&L).

| Bid Amount | Bid Price | Ask Price | Ask Amount |
|---|---|---|---|
|  |  | 8761.89 | 1200 |
|  |  | 8761.87 | 1652 |
|  |  | 8761.84 | 1856 |
|  |  | 8761.63 | 2952 |
|  |  | 8761.41 | 3000 |
| 2500 | 8761.40 |  |  |
| 2300 | 8761.20 |  |  |
| 2200 | 8761.15 |  |  |
| 1800 | 8761.01 |  |  |
| 1750 | 8760.98 |  |  |

**Fig 1. Orderbook snapshot for btc-usd.**

with purchase orders and an ask list with sell orders, with orders on either list quoting both a quantity of assets and the price at which the buyer or seller, respectively, are willing to trade them. The difference between the lowest ask price and highest bid price for an asset is called the spread. Fig 1 shows a section of an orderbook where bid quotes (left side) and ask quotes (right side) meet across a spread of 0.01 (8761.41 − 8761.40). Market makers place both bid and ask quotes in the orderbook, thus generating demand for and supply of the asset for prospective sellers and buyers, respectively.

The cumulative profit (or loss) resulting from a market maker's operations comes from the successive execution of trades on both sides of the spread. This profit from the spread is endangered when the market maker's buy and sell operations are not balanced overall in volume, since this will increase the dealer's asset inventory. The larger the inventory is, be it positive (long stock) or negative (short stock), the higher the holder's exposure to market movements. Hence, market makers try to minimize risk by keeping their inventory as close to zero as possible. Market makers tend to do better in mean-reverting environments, whereas market momentum, in either direction, hurts their performance.

Inventory management is therefore central to market making strategies (see section 2 for an overview of these), and particularly important in high-frequency algorithmic trading. In an influential paper [2], Avellaneda and Stoikov expounded a strategy addressing market maker inventory risk. Essentially, the Avellaneda-Stoikov (AS) algorithm derives optimal bid and ask quotes for the market maker to place at any given moment, by leveraging a statistical model of the expected sizes and arrival times of market orders, given certain market parameters and a specified degree of risk aversion in the market maker's quoting policy. The optimal bid and ask quotes are obtained from a set of formulas built around these parameters. These formulas prescribe the AS strategy for placing limit orders. The rationale behind the strategy is, in Avellaneda and Stoikov's words, to perform a '*balancing act between the dealer's personal risk considerations and the market environment*' [ibid.].

The AS algorithm is static in its reliance on analytical formulas to generate bid and ask quotes based on the real-time input values for the market mid-price of the security and the current stock inventory held by the market maker. These formulas (as we will see in section 2) have fixed parameters to model the market maker's aversion to risk and the statistical properties of market orders.

In this paper we present a limit order placement strategy based on a well-known reinforcement learning (RL) algorithm. The peculiarity of our approach is that, rather than relying on

this RL algorithm directly to determine what limit orders to place (as all other machine learning-based methods in the literature do, to our knowledge), we still use the AS algorithm to determine bid and ask quotes. We use the RL algorithm to modify the risk aversion parameter and to skew the AS quotes based on a characterization of the latest steps of market activity. Another distinctive feature of our work is the use of a genetic algorithm to determine the parameters of the AS formulas, which we use as a benchmark, to offer a fairer performance comparison to our RL algorithm.

The paper is organized as follows. The Avellaneda-Stoikov procedure underpinning the market-making actions in the models under discussion is explained in Section 2. Section 3 provides an overview of reinforcement learning and its uses in algorithmic trading. The deep reinforcement learning models (Alpha-AS-1 and Alpha-AS-2) developed to work with the Avellaneda-Stoikov algorithm are presented in detail in Section 4, together with an Avellaneda-Stoikov model (Gen-AS) without RL with parameters obtained with a genetic algorithm. Section 5 describes the experimental setup for backtests that were performed on our RL models, the Gen-AS model and two simple baselines. The results obtained from these tests are discussed in Section 6. The concluding Section 7 summarises the approach and findings, and outlines ideas for model improvement.

## 2 Background: The Avellaneda-Stoikov procedure

In 2008, Avellaneda and Stoikov published a procedure to obtain bid and ask quotes for high-frequency market-making trading [2, 3]. The successive orders generated by this procedure maximize the expected exponential utility of the trader's profit and loss (P&L) profile at a future time, $T$ (usually, the daily closing time for trade), for a given level of agent inventory risk aversion. Intuitively, the underlying idea is, first, to adjust the market mid-price taking into account the size of the stock inventory held by the agent, the market volatility and the time remaining until $T$, these all being factors affecting inventory risk, and adjusting also according to the agent's sensitivity to this risk (i.e., the risk aversion, which is assumed to be constant); then the agent's bid and ask quotes are set around this adjusted mid-price, called the *reservation* price, at a distance at which their probability of execution is optimal, i.e., it leads, through repeated application, to the maximization of profit at time $T$.

The procedure, therefore, has two steps, which are applied at each time increment as follows.

1. Set the reservation price, $r$:

$$r(t_j) = p^m(t_j) - I(t_j)\gamma\sigma^2(T - t_j) \tag{1}$$

where $t_j$ is the current time upon arrival of the j[th] market tick, $p^m(t_j)$ is the current market mid-price, $I(t_j)$ is the current size of the inventory held, $\gamma$ is a constant that models the agent's risk aversion, and $\sigma^2$ is the variance of the market midprice, a measure of volatility. We should note that $r$ is actually the average of a bid indifference price ($r^b$) and an ask indifference price ($r^a$), which are defined mathematically to be, respectively, the stock bid and ask quote prices at which the agent's expected P&L utility will be the same whether a stock is bought or not (for the bid indifference price) or sold or not (in the case of the ask indifference price), thus making the agent indifferent to placing orders at these prices. This consideration makes $r^b$ and $r^a$ (rather than s) reasonable reference prices around which to construct the market maker's spread. Avellaneda and Stoikov define $r^b$ and $r^a$, however, for a passive agent with no orders in the limit order book. In practice, as Avellaneda and Stoikov did in their original paper, when an agent is running and placing orders both $r^b$ and ra $r^a$ are approximated by the average of the two, $r$ [2].

2. Calculate the spread $(p^a - p^b)$:

$$\delta^a(t) = \frac{1}{2}\gamma\sigma^2\left(T - t_j\right) + \frac{1}{\gamma}ln\left(1 + \frac{\gamma}{k^a}\right) \tag{2}$$

$$\delta^b(t) = \frac{1}{2}\gamma\sigma^2\left(T - t_j\right) + \frac{1}{\gamma}ln\left(1 + \frac{\gamma}{k^b}\right) \tag{3}$$

Here, $\delta$ is the distance from the reservation price, $r$, at which bid and ask quotes will be generated, on either side of $r$. The $k$ parameter models order book liquidity, with larger values corresponding to higher trading intensity. For a specific time interval, $\pi_n = t_n - t_{n-1}$, $k$ can be estimated as done in [3]:

$$k_n^a = \frac{\lambda_n^a \cdot \lambda_{n-1}^a}{\lambda_n^a - \lambda_{n-1}^a} \tag{4}$$

$$k_n^b = \frac{\lambda_n^b \cdot \lambda_{n-1}^b}{\lambda_n^b - \lambda_{n-1}^b} \tag{5}$$

where $\lambda_n^a$ and $\lambda_n^b$ are the orderbook update arrival rates on the ask and bid sides, respectively, in the time interval $\pi_n = t_n - t_{n-1}$. Note that this approach, following Aldridge's [3], allows us to estimate the $k$ parameters simply by counting the order arrivals in each time interval, $\pi_n$. No further parameter is needed to characterise the asset's liquidity (such as $A$, if we were to model order arrival rates by the exponential law $\lambda(\delta) = Ae^{-k\delta}$. as in [2, 4]).

We apply a symmetric spread around the reservation price. Hence, we set the ask price, $p^a$, and the bid price, $p^b$, as:

$$p^a = r + \delta^a \tag{6}$$

$$p^b = r - \delta^b \tag{7}$$

where all terms are evaluated at time $t_j$.

From these equations we see that the larger a positive inventory held ($I$) is, the lower the reservation price drops below the market mid-price. This will skew the ask and bid prices downward with respect to the market mid-price, making selling stock more likely than buying it. Conversely, the greater a negative inventory is, the more skewed the ask and bid prices will be above the market mid-price, thus increasing the probability of buying stock and decreasing that of selling it. The combined effect is to pull the inventory back toward zero, and hence also the risk inherent to holding it. The expression for $r$ (Eq (1)) ensures the AS strategy is sensitive to price volatility ($\sigma$), by widening the spread when volatility is high. Thus, order placement is more cautious when the market is more unpredictable, which reduces risk. Inventory risk also diminishes as trade draws closer to termination time T, since the market has less time in which to move. This is reflected in the AS procedure by the convergence of $r(t_j)$ to $p^m(t_j)$ (Eq (1)) and the narrowing of the spread (Eq (2) as $t_j \rightarrow T$. We also observe that the difference between the reservation price and the market mid-price is proportional to the agent's risk aversion. As regards the market liquidity parameter, $k$, a low (high) value models a market with low (high) trading intensity. With fewer market updates, placing quotes entails a greater inventory risk; conversely, high market intensity reduces inventory risk, as both buy and sell orders are more likely to be executed, keeping the inventory balanced. The risk management associated with $k$

is addressed by Eq (2), by making the spread increase as $k$ decreases (thus further decreasing the probability that the orders placed will be executed within a given time interval), and vice versa.

The models underlying the AS procedure, as well as its implementations in practice, rely on certain assumptions. Statistical assumptions are made in deriving the formulas that solve the P&L maximization problem. First, it is assumed that the agent's orders are executed at a Poisson rate which decreases as the spread increases (i.e., the farther away from the market midprice an order is placed at, the more time should tend to elapse before it is executed); second, the arrival frequency of market updates is assumed to be constant; and third, the distribution of the size of these orders, as well as their market impact (which is an estimation of the price change a buy or sell order of a certain magnitude can affect on the arrival rates of market orders), are taken to follow some given law [2]. For instance, Avellaneda and Stoikov [2] (ibid.) illustrate their method using a power law to model market order size distribution and a logarithmic law to model the market impact of orders. Furthermore, as already mentioned, the agent's risk aversion ($\gamma$) is modelled as constant in the AS formulas. Finally, as noted above, implementations of the AS procedure typically use the reservation price ($r$) as an approximation for both the bid and ask indifference prices.

The AS model generates bid and ask quotes that aim to maximize the market maker's P&L profile for a given level of inventory risk the agent is willing to take, relying on certain assumptions regarding the microstructure and stochastic dynamics of the market. Extensions to the AS model have been proposed, most notably the Guéant-Lehalle-Fernandez-Tapia approximation [5], and in a recent variation of it by Bergault et al. [6], which are currently used by major market making agents. Nevertheless, in practice, deviations from the model scenarios are to be expected. Under real trading conditions, therefore, there is room for improvement upon the orders generated by the closed-form AS model and its variants.

One way to improve the performance of an AS model is by tweaking the values of its constants to fit more closely the trading environment in which it is operating. In section 4.2, we describe our approach of using genetic algorithms to optimize the values of the AS model constants using trading data from the market we will operate in. Alternatively, we can resort to machine learning algorithms to adjust the AS model constants and/or its output ask and bid prices dynamically, as patterns found in market-related data evolve. To this approach, more specifically one based on deep reinforcement learning, we turn to next.

## 3 Related work on machine learning in trading

One of the most active areas of research in algorithmic trading is, broadly, the application of machine learning algorithms to derive trading decisions based on underlying trends in the volatile and hard to predict activity of securities markets. Machine learning (ML) is being applied to time series prediction (for instance, of next-day prices [7, 8]); risk management (e.g., in [9] a ML model is substituted for the commonly used Principal Components Analysis approach), and the improvement or discovery of factors in factor investing [10–13]. Machine learning approaches have been explored to obtain dynamic limit order placement strategies that attempt to adapt in real time to changing market conditions. Collado and Creamer [14] performed time series forecasting using dynamic programming; deep neural networks have found undervalued equities [15]; reinforcement learning has been used successfully in execution algorithms to lessen market impact [16], as well as to hedge a derivatives portfolio, simulating liquidity, market impact and transaction costs by learning from a nonlinear environment [17]. As regards market making, the AS algorithm, or versions of it [3], have been used as benchmarks against which to measure the improved performance of the machine learning

algorithms proposed, either working with simulated data [18] or in backtests [8] with real data. The literature on machine learning approaches to market making is extensive.

We now turn to uses in algorithmic trading of a specific branch of machine learning: reinforcement learning.

### 3.1 A brief overview of the reinforcement learning paradigm

A branch of machine learning that has drawn particularly strong attention from the field of algorithmic trading is *reinforcement learning* (RL), already a feature in some of the aforementioned work. Through interaction with its environment, a reinforcement learning algorithm learns a *policy* to guide its actions, with the goal of optimizing a reward that it obtains by said interaction. The policy determines what action it is best to perform in a given situation, as part of a sequence of actions, such that when the sequence terminates the cumulative reward is maximized. The RL paradigm is built upon the following elements (Fig 2): an agent with a quantifiable goal acts upon its environment according to information it receives from the environment regarding both its state (which may have changed at least partly as a consequence of the agent's previous actions) and the goal-relevant consequences of the agent's previous actions, quantified as a cumulative reward to be maximized.

Applied to market making, the goal of the RL agent is to maximize the expected P&L profile utility at some future time, T. In each action-reward cycle the agent reads the current state of the order book (its environment): the market mid-price and details of the order book microstructure. As its actions in pursuit of its goal, the agent places buy and sell orders in the order book. From these orders it obtains a reward: a profit or a loss. The reward, together with the new state of the order book (which will have changed through the accumulated actions of all the agents operating in the market), are taken into account by the agent to decide its actions in the next cycle.

The interplay between the agent and its environment can be modelled as a Markov Decision Process (MDP), which defines:

- A state space ($\mathcal{S}$): the set of states the environment can be in.

- An action space ($\mathcal{A}$): the set of actions available to the agent.

- A transition function ($\mathcal{T}$) that specifies the probabilities of transitioning from a given state to another when a given action is executed.

- A reward function ($\mathcal{R}$), that associates a reward with each transition.

- A discount factor ($\gamma$) by which future rewards are given less weight than more immediate ones when estimating the value of an action (an action's value is its relative worth in terms of the maximization of the cumulative reward at termination time).

Typically, in the beginning the agent does not know the transition and reward functions. It must explore actions in different states and record how the environment responds in each case. Through repeated *exploration* the agent gradually learns the relationships between states,

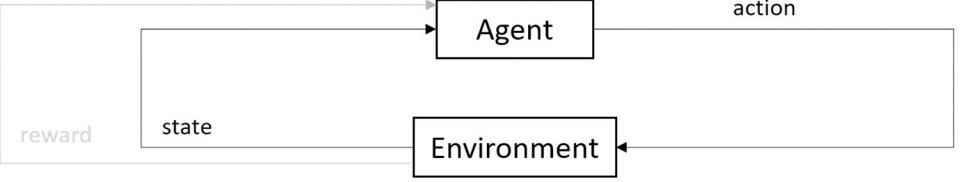

**Fig 2. The reinforcement learning paradigm.** (Adapted from [Sutton & Barto] [19]).

actions and rewards. It can then start *exploiting* this knowledge to apply an action selection policy that takes it closer to achieving its reward maximization goal.

A wide variety of RL techniques have been developed to allow the agent to learn from the rewards it receives as a result of its successive interactions with the environment. Deep reinforcement learning (DRL) is a subfamily of RL algorithms based on artificial neural networks, that in recent years have surpassed human ability to solve problems that were previously unassailable via machine learning approaches, due primarily to the vast decision space to be explored. A notable example is Google's AlphaGo project [20], in which a deep reinforcement learning algorithm was given the rules of the game of Go, and it then taught itself to play so well that it defeated the human world champion. AlphaGo learned by playing against itself many times, registering the moves that were more likely to lead to victory in any given situation, thus gradually improving its overall strategies. The same concept has been applied to train a machine to play Atari video games competently, feeding a convolutional neural network with the pixel values of successive screen stills from the games [21].

## 3.2 Reinforcement learning in algorithmic trading

These successes with games have attracted attention from other areas, including finance and algorithmic trading. The large amount of data available in these fields makes it possible to run reliable environment simulations with which to train DRL algorithms. DRL is widely used in the algorithmic trading world, primarily to determine the best action (buy or sell) to take in trading by candles, by predicting what the market is going to do. For instance, Lee and Jangmin [22] used Q-learning with two pairs of agents cooperating to predict market trends (through two "signal" agents, one on the buy side and one on the sell side) and determine a trading strategy (through a buy "order" agent and a sell "order" agent). RL has also been used to dose buying and selling optimally, in order to reduce the market impact of high-volume trades which would damage the trader's returns [16].

In most of the many applications of RL to trading, the purpose is to create or to clear an asset inventory. The more specific context of market making has its own peculiarities. DRL has been used generally to determine the actions of placing bid and ask quotes directly [23–26], that is, to decide when to place a buy or sell order and at what price, without relying on the AS model. Guéant and Manziuk [27] have proposed a DRL-based approach to deriving approximations to the optimal bid and ask quotes for P&L maximization across a large number assets (corporate bonds), overcoming the insurmountable obstacle faced by analytical approaches to solving the high-dimensional systems of equations involved (the familiar *curse of dimensionality*). Spooner [24] proposed a RL system in which the agent could choose from a set of 10 spread sizes on the buy and the sell side, with the asymmetric dampened P&L as the reward function (instead of the plain P&L). Combining a deep Q-network (DQN) (see Section 4.1.7) with a convolutional neural network (CNN), Juchli [23] achieved improved performance over previous benchmarks. Kumar [26], who uses Spooner's RL algorithm as a benchmark, proposes using deep recurrent Q-networks (DRQN) as an improved alternative to DQNs for a time-series data environment such as trading. Gašperov and Konstanjčar [25] tackle the problem be means of an ensemble of supervised learning models that provide predictive buy/sell signals as inputs to a DRL network trained with a genetic algorithm. The same authors have recently explored the use of a soft actor-critic RL algorithm in market making, to obtain a continuous action space of spread values [28]. Comprehensive examinations of the use of RL in market making can be found in Gašperov et al. [29] and Patel [30].

What is common to all the above approaches is their reliance on learning agents to place buy and sell orders directly. That is, these agents decide the bid and ask prices of their

orderbook quotes at each execution step. The main contribution we present in this paper resides in delegating the quoting to the mathematically optimal Avellaneda-Stoikov procedure. What our RL algorithm determines are, as we shall see shortly, the values of the main parameters of the AS model. It is then the latter that calculates the optimal bid and ask prices at each step.

## 4 Models

The RL agents (Alpha-AS) developed to use the Avellaneda-Stoikov equations to determine their actions (the bid and ask prices place in the orderbook) are described in Section 4.1. An agent that simply applies the Avellaneda-Stoikov procedure with fixed parameters (Gen-AS), and the genetic algorithm to obtain said parameters, are presented in Section 4.2.

### 4.1 The Alpha-AS model

Hasselt, Guez and Silver [31] developed an algorithm they called double DQN. Double DQN is a deep RL approach, more specifically deep Q-learning, that relies on two neural networks, as we shall see shortly (in Section 4.1.7). In this paper we present a double DQN applied to the market-making decision process.

**4.1.1 The concept.** The usual approach in algorithmic trading research is to use machine learning algorithms to determine the buy and sell orders directly. These orders are the output actions of each execution cycle. In contrast, we propose maintaining the Avellaneda-Stoikov procedure as the basis upon which to determine the orders to be placed. We use a reinforcement learning algorithm, a double DQN, to adjust, at each trading step, the values of the parameters that are modelled as constants in the AS procedure. The actions performed by our RL agent are the setting of the AS parameter values for the next execution cycle. With these values, the AS model will determine the next reservation price and spread to use for the following orders. In other words, we do not entrust the entire order placement decision process to the RL algorithm, learning through blind trial and error. Rather, taking inspiration from Teleña [32], we mediate the order placement decisions through the AS model (our "avatar", taking the term from [32]), leveraging its ability to provide quotes that maximize profit in the ideal case. In humble homage to Google's AlphaGo programme, we will refer to our double DQN algorithm as *Alpha-Avellaneda-Stoikov (Alpha-AS)*.

**4.1.2 Background.** double DQN [31] builds on Deep Q-learning, which in turn is based on the Q-learning algorithm.

*Q-learning.* Q-learning is an early RL algorithm for Markov decision processes, developed from Bellman's recursive Q-value iteration algorithm [33] for estimating, for each possible state-action pair, $(s, a)$, the sum of future rewards (the Q-value) that will be accrued by choosing that action from that state, assuming all future choices will be optimal (i.e., assuming the action chosen in any given state arrived at in future steps will be the one with the highest Q-value). The Q-value iteration algorithm assumes that both the transition probability matrix and the reward matrix are known.

The Q-learning algorithm, on the other hand, estimates the Q-values–the $Q_{s,a}$ matrix–with no prior knowledge of the transition probabilities or of the rewards. At each iteration, $i$, the values in the $Q_{s,a}$ matrix are updated taking into account the observed reward obtained from the latest state-action pair, as described by the following equation [19]:

$$Q_{i+1}(s, a) = Q_i(s, a) + \alpha[R(s, a) + \gamma_d \max_{a'} Q_i(s', a') - Q_i(s, a)] \qquad (8)$$

where:

- $R(s, a)$ is the latest reward obtained from state $s$ by taking action $a$.

- $s'$ is the state the MDP has transitioned to when taking action $a$ from state $s$, to which it arrived at the previous iteration.

- $\max_{a'} Q_i(s', a')$ is the highest Q-value estimate (corresponding to action $a'$) already stored for the new state, $s'$, from among those of all the state-action pairs available in state $s'$.

- $\gamma_d$ is a discount factor ($\gamma_d \in [0, 1]$) by which future expected rewards are given less weight in the current Q-value than the latest observed reward. ($\gamma_d$ is usually denoted simply as $\gamma$, but in this paper we reserve the latter to denote the risk aversion parameter of the AS procedure).

- $\alpha$ is the learning rate ($\alpha \in [0, 1]$), which reduces to a fraction the amount of change that is applied to $Q_i(s, a)$ from the observation of the latest reward and the expectation of optimal future rewards. This limits the influence of a single observation on the Q-value to which it contributes.

- $Q_i(s, a)$ is known as the *prediction* Q-value.

- The $[R(s, a) + \gamma_d \ \max_{a'} Q_i(s', a')]$ term is referred to as the *target* Q-value.

The algorithm combines an exploration strategy to reach an increasing number of states and try the different available actions to obtain examples with which to estimate the optimal Q-value for each state-action pair, with an exploitation policy that uses the obtained Q-value estimates to select, at each step, an action with the aim of maximising the total future reward. Balancing exploration and exploitation advantageously is a central challenge in RL.

*Deep Q-learning.* For even moderately large numbers of states and actions, let alone when the state space is practically continuous (which is the case presented in this paper), it becomes computationally prohibitive to maintain a $Q_{s,a}$ matrix and iteratively to get the values contained in it to converge to the optimal Q-value estimates. To overcome this problem, a deep Q-network (DQN) approximates the $Q_{s,a}$ matrix using a deep neural network. The DQN computes an approximation of the Q-values as a function, $Q(s, a, \boldsymbol{\theta})$, of a parameter vector, $\boldsymbol{\theta}$, of tractable size. To train a DQN is to let it evolve the values of these internal parameters based on the agent's experiences acting in its environment, so that the value function approximated with them maps the input state to Q-values that increasingly approach the optimal Q-values for that state. There are various methods to achieve this, a particularly common one being gradient descent.

The general architecture of a DQN is as follows:

- Input layer: for an MDP with a state space determined by the combinations of values that a set of variables may take (as is the case of the Alpha-AS model we describe in Section 4.1), the input layer of a DQN will typically have one neuron for each input variable.

- Output layer: one neuron per action available to the agent. Each output neuron will give the new Q-value estimate for the corresponding action, after processing the latest observation vector input to the network.

- One or several hidden layers, the structure of which can vary greatly from system to system.

Thus, the DQN approximates a Q-learning function by outputting for each input state, $s$, a vector of Q-values, which is equivalent (approximately) to checking the row for $s$ in a $Q_{s,a}$ matrix to obtain the Q-value for each action from that state.

A second problem with Q-learning is that performance can be unstable. Increasing the number of training experiences may result in a decrease in performance; effectively, a loss of

learning. To improve stability, a DQN stores its experiences in a *replay buffer*, in terms of the value function given by Eq (8), where now the Q-value estimates are not stored in a matrix but obtained as the outputs of the neural network, given the current state as its input. A *policy function* is then applied to decide the next action. A common function is an *ε-greedy* policy that balances exploration and exploitation, randomly exploring new actions from the current state with probability ε, and otherwise (with probability 1−ε) exploiting the knowledge contained in the neural network by performing the action it recommends as its output given the current state. approximate Q-values stored for the state. The DQN then learns periodically, with batches of random samples drawn from the replay buffer, thus covering more of the state space, which accelerates the learning while diminishing the influence of single or of correlated experiences on the learning process.

*Double DQNs.* Double DQNs [31] represent a further improvement on DQN algorithms, in terms of training stability and performance. Using a single DQN to determine both the prediction and the target Q-values results in random overestimations of the latter values (ibid.). To address this problem, as their name suggests, double DQNs rely on two DQNs: a *prediction* DQN and a *target* DQN. The prediction DQN works as the DQNs discussed so far, but with target values set by the target DQN. The target DQN is structurally identical to the prediction DQN. However, the parameters of the target DQN, are updated only once every given number of training iterations, simply by copying the parameters of the prediction DQN, which in the meantime will have been modified by exposure to new experiences.

Both the prediction DQN and the target DQN are used to solve the Bellman Eq (8) and obtain $Q_{i+1}(s, a)$ at each iteration. Once again, the prediction DQN provides $Q_i(s, a)$ while the target DQN gives $\max_{a'} Q_i(s', a')$. We can now write the value function of the double DQN as:

$$Q_{i+1}(s, a) = PredictionDQN_i(s, a) + \alpha[R(s, a) + \gamma_d \max_{a'} TargetDQN_i(s', a')$$
$$- PredictionDQN_i(s, a)] \tag{9}$$

The exploitation policy function chooses the next action, $a'$, as that which maximises the output of the *prediction* DQN:

$$a' = \max_a PredictionDQN_i(s, a) \tag{10}$$

We model the market-agent interplay as a Markov Decision Process with initially unknown state transition probabilities and rewards.

**4.1.3 Time step ($\Delta \tau = \tau_{i+1} - \tau_i$).**   The time step of the action-reward cycle is 5 seconds of trading time. The agent is going to repeat the chosen action at every orderbook tick that occurs throughout the time step. It will accumulate the reward obtained through the repeated application of the action during this time. As we shall see shortly, the actions specify two things: the risk aversion parameter in the AS formulas and a skew applied to the prices returned by the formulas. Repeating the action simply means setting these (and only these) two parameters, risk aversion and skew, to the same values for the duration of the 5-second time window. With these parameters thus updated every 5 seconds, fresh bid and ask prices are obtained at every tick, with the latest market information, through the application of the AS formulas.

**4.1.4 States (S).**   We characterize the Alpha-AS agent and its environment (the market) through a set of state-defining features. We divide the feature set conceptually into two subsets (adapting the nomenclature in [19]):

- *Private indicators*, consisting of features describing the state of the agent.

- *Market indicators*, consisting of features describing the state of the environment.

The features will reflect inventory levels, market prices and other indicators derived from these. For each indicator considered, we define $N$ features holding the bucket identifiers corresponding to the current value of the feature (denoted with the suffix $X = 0$) and its values for the previous $N$-1 ticks or candles (denoted with the suffixes $X = 1, 2, \ldots N$-1, respectively). In other words, the agent will use a horizon of the $N$ latest values of each feature, that is, the values the feature has taken in the last $N$ ticks (orders entered in the order book, as well as cancellations). That is, the values for each feature are stored in a circular First-In First-Out queue of size $N$, with overwriting. Should more than $N$ ticks occur in the 5-second window, only the last $N$ will be in the queue for consideration when determining the actions for the next 5-second time step; conversely, in the rare event that fewer than $N$ ticks occur in a time step, some values from the previous time step will still be in the queue, and thus taken into account again. The value of $N$ will vary for different features, as specified below, and in the case of the market candle indicators it refers to candles, not ticks. In each case, a value of $N$ was chosen large enough to provide the agent with a sufficiently rich state space from which to learn, while also small enough that training demands a manageable amount of time and resources.

The feature quantities are very fine-grained. To derive a manageable number of states from the combinations of all possible feature values, we defined for each a set of value buckets, as follows:

a. The feature values are discretised by rounding to a number of decimals, $d$, specific to each type of feature ($d = 3$ or $7$).

b. The ranges of possible values of the features that are defined in relation to the market midprice, are truncated to the interval $[-1, 1]$ (i.e., if a value exceeds 1 in magnitude, it is set to 1 if it is positive or -1 if negative).

Together, a) and b) result in a set of $2 \times 10^d$ contiguous buckets of width $10^{-d}$, ranging from $-1$ to $1$, for each of the features defined in relative terms. Approximately 80% of their values lie in the interval $[-0.1, 0.1]$, while roughly 10% lie outside the $[-1, 1]$ interval. Values that are very large can have a disproportionately strong influence on the statistical normalisation of all values prior to being inputted to the neural networks. By trimming the values to the $[-1, 1]$ interval we limit the influence of this minority of values. The price to pay is a diminished nuance in the learning from very large values, while retaining a higher sensitivity for the majority, which are much smaller. By truncating we also limit potentially spurious effects of noise in the data, which can be particularly acute with cryptocurrency data.

A full set of buckets, one for each *selected* feature, is associated with a state. That is, the agent designates the same state for a particular combination of feature buckets, regardless of the precise values obtained for each feature (as long as they fall in the corresponding state-defining buckets).

To further reduce the number of states considered by the RL agent and so lessen the familiar *curse of dimensionality* [19], taking inspiration from [34], we selected the top 35% from the complete set of defined features, as determined by their scores on feature importance metrics for random forest classifiers (see *Feature selection*, below).

*Indicators and feature definition*: *Private indicators*. The agent describes itself by the amount of inventory it holds and the reward it receives after performing actions. For each indicator the agent defines 5 features ($N = 5$), to hold its current ($X = 0$) and its 4 previous values. The values are rounded to 3 decimals ($d = 3$). (This results in a total of 2000 buckets of size 0.001, from values -1 to 1, with the lowest bucket being assigned to any feature value -0.999 or lower, and the highest bucket to any value above 0.999, however large.)

The features are as follows (with $0 \leq X \leq 4$):

- **inventory_X**: inventory level, divided by the inventory quantity quoted.

- **score_X**: the cumulative Asymmetric dampened P&L (see the Reward specification below) obtained so far in the current day of trading, divided by the inventory quantity quoted.

As we shall see shortly, the reward function is the Asymmetric dampened P&L obtained in the current 5-second time step. In contrast, the total P&L accrued so far in the day is what has been added to the agent's state space, since it is reasonable for this value to affect the agent's assessment of risk, and hence also how it manipulates its risk aversion as part of its ongoing actions.

*Market indicators.* The Alpha-AS agent describes its environment through two sets of market indicators: market tick indicators and market candle indicators. Market tick indicators are updated every time a new order appears in the orderbook; market candle indicators are updated at regular time intervals, and they reflect the overall market change in the last interval (which may have seen any number of ticks). We set the candle duration to 1 minute of trading.

*Market tick indicators.* For each market tick indicator the agent defines 10 features ($N = 10$), to hold its current ($X = 0$) and its 9 previous values. The values are rounded to 7 decimals ($d = 7$, yielding $2 \cdot 10^7$ buckets). All price-related tick features (but not the quantity-related features) are given as their difference to the current mid-price (the midpoint between the best ask price and best bid price in the orderbook).

The market tick features are the following (with $0 \leq X \leq 9$):

- **ask_price_X**: the best ask price.

- **ask_qty_X**: the quantity of assets available in the market at the best ask price.

- **bid_price_X**: the best bid price.

- **bid_qty_X**: the quantity of assets that are currently being bid for in the market at the best bid price.

- **spread_X**: the best ask price minus the best bid price in the orderbook.

- **last_close_price_X**: the price at which the latest trade was executed in the market.

- **microprice_X**: the orderbook microprice [35], as defined by Eq (11).

- **imbalance_X**: the orderbook imbalance, as defined by Eq (12).

$$microprice = \frac{AskQty_0 \cdot AskPrice_0 + BidQty_0 \cdot BidPrice_0}{AskQty_0 + BidQty_0} \tag{11}$$

where the 0 subscript denotes the best orderbook price level on the ask and on the bid side, i.e., the price levels of the lowest ask and of the highest bid, respectively.

$$imbalance = \frac{\sum_{level=0}^{maxdepth} BidQty_{level} - \sum_{level=0}^{maxdepth} AskQty_{level}}{\sum_{level=0}^{maxdepth} BidQty_{level} + \sum_{level=0}^{maxdepth} AskQty_{level}} \tag{12}$$

where $\sum_{level=0}^{maxdepth}(\cdot)$ is the sum of the corresponding quantity over all of the orderbook levels (best to worse price).

*Market candle indicators.* For each market candle indicator the agent defines 3 features ($N = 3$), to hold its value for the current candle ($X = 0$) and the 2 previous candles ($X = 1$ and $X = 2$, respectively). The values are rounded to 3 decimals ($d = 3$). The market candle features

are normalized by the open mid-price (i.e., the mid-price at the start of the candle). They are the following (with $0 \leq X \leq 2$):

- **close_X**: the last mid-price in candle **X** (divided by the open mid-price for the candle).

- **low_X**: the lowest mid-price in candle **X** (divided by the open mid-price for the candle).

- **high_X**: the highest mid-price in candle **X** (divided by the open mid-price for the candle).

- **ma**: the mean of the 3 **close_X** values.

- **std**: the standard deviation of the 3 **close_X** values.

- **min**: the lowest mid-price in the latest 3 candles (i.e., the lowest of the **low_X** values).

- **max**: the highest mid-price in the latest 3 candles (i.e., the highest of the **high_X** values).

*Feature selection.* Reducing the number of features considered by the RL agent in turn dramatically reduces the number of states. This helps the algorithm learn and improves its performance by reducing latency and memory requirements.

Following the approach in López de Prado [34], where random forests are applied to an automatic classification task, we performed a selection from among our market features (tick and candle), based on a random forest classifier. We did not include the 10 private features (the 5 latest inventory levels and 5 latest rewards) in the feature selection process, as we want our algorithms always to take these agent-related (as opposed to environment-related) values into account.

The target for the random forest classifier is simply the sign of the difference in mid-prices at the start and the end of each 5-second timestep. That is, classification is based on whether the mid-price went up or down in each timestep. The labels are the state features themselves.

Three standard feature importance metrics were used to select the 35% of all market features that had the greatest impact on the output of the agent's reward function (we relied on MLfinlab's python implementation to calculate these three metrics [36]:

- *Mean decrease impurity* (MDI), a feature-specific measure of the mean reduction of weighted impurity over all the nodes in the tree ensemble that partition the data samples according to the values of that feature [34]. We used entropy as the impurity metric. The 8.75% highest-scoring features on MDI were retained.

- *Mean decrease accuracy* (MDA), a feature-specific estimate of average decrease in classification accuracy, across the tree ensemble, when the values of the feature are permuted between the samples of a test input set [34]. To obtain MDA values we applied a random forest classifier to the dataset split in 4 folds. The 8.75% highest-scoring features on MDA were retained.

- *Single feature importance* (SFI), an out-of-sample estimator of the individual importance of each feature, that avoids the substitution effect found with MDI and MDA (important features are ignored when highly correlated with other important features). The 17.5% highest-scoring features on SFI were retained.

The data on which the metrics for our market features were calculated correspond to one full day of trading ($7^{\text{th}}$ December 2020). The selection of features based on these three metrics reduced their number from 112 to 22 (there was some overlap in the features selected by the different metrics). The features retained by each importance indicator are shown in Table 1.

The two most important features for all three methods are the latest bid and ask quantities in the orderbook (*ask_qty_0* and *ask_qty_0*), followed closely by the bid and ask quantities immediately prior to the latest orderbook update (*ask_qty_1* and *ask_qty_1*) and the latest best

**Table 1. Features ordered by importance according to the metrics MDI, MDA and SFI.**

| Rank | MDI | MDA | SFI |
|---|---|---|---|
| 1 | **bid_qty_0** | bid_qty_0 | ask_qty_0 |
| 2 | **ask_qty_0** | ask_qty_0 | bid_qty_0 |
| 3 | **ask_qty_1** | microprice_0 | ask_qty_1 |
| 4 | **microprice_0** | ask_price_0 | bid_qty_1 |
| 5 | **ask_qty_3** | bid_qty_1 | ask_price_0 |
| 6 | **bid_price_0** | bid_price_0 | spread_0 |
| 7 | **ask_price_0** | **spread_0** | bid_price_0 |
| 8 | **bid_qty_1** | **ask_qty_2** | **low_0** |
| 9 | **last_close_price_4** | **midprice_8** | **microprice_8** |
| 10 | | | **spread_8** |
| 11 | | | **ask_price_8** |
| 12 | | | **bid_price_8** |
| 13 | | | **ask_price_4** |
| 14 | | | **spread_4** |
| 15 | | | **bid_price_4** |
| 16 | | | ask_qty_2 |
| 17 | | | **bid_qty_2** |
| 18 | | | **high_4** |

(The first appearance of a feature, from left to right, is shown in bold).

ask and bid prices (*ask_price_0* and *bid_price_0*). There is a general predominance of features corresponding to the latest orderbook movements (i.e., those denominated with low numerals, primarily 0 and 1). This may be a consequence of the markedly stochastic nature of market behaviour, which tends to limit the predictive power of any feature to proximate market movements. Hence the heightened importance of the latest market tick when determining the following action, even if the actor is beholden to take the same action repeatedly during the next 5 seconds, only re-evaluating the action-determining market features after said period has elapsed. Nevertheless, the prices 4 and 8 orderbook movements prior the action setting instant also make fairly a strong appearance in the importance indicator lists (particularly for SFI), suggesting the existence of slightly longer-term predictive component that may be tapped into profitably.

The total number of features retained to define the states for our agents is therefore 32: the 10 private features and these 22 market features.

**4.1.5 Actions ($A$).** The actions taken by the Alpha-AS agent rely on the ask and bid prices given by the Avellaneda-Stoikov procedure. As its action, to repeat for the duration of the time step, the agent chooses the values of two parameters to apply to this procedure: risk aversion ($\gamma$) and skew (which alters the ask and bid prices obtained with the AS method). For each of these parameters the values are chosen from a finite set, as follows:

- **Risk aversion ($\gamma$)**: a parameter of the AS model itself, as discussed in Section 2. At each time step, before applying the AS procedure to obtain ask and bid prices (Eqs (1), (2) and (3)), the agent selects a value for $\gamma$ from the set {0.01, 0.1, 0.2, 0.9}. We restrict the agent's choice to these values so that it usually behaves with high risk aversion (high values of $\gamma$) but is also able to switch to a more aggressive, low risk aversion strategy (setting $\gamma = 0.01$) when it deems the conditions so require.

- **Skew**: after a bid and ask price are obtained by the AS procedure, the agent modifies them by a fraction given by the skew. The modified formulas for the ask and bid price are:

$$p^a = (r + \delta_a)(1 + Skew) \tag{13}$$

$$p^b = (r - \delta_b)(1 + Skew) \tag{14}$$

Where the value for the Skew is chosen from the set {−0.1, −0.05, 0, 0.05, 0.1}. Therefore, by choosing a *Skew* value the Alpha-AS agent can shift the output price upwards or downwards by up to 10%.

The combination of the choice of one from among four available values for $\gamma$, with the choice of one among five values for the skew, consequently results in 20 possible actions for the agent to choose from, each being a distinct ($\gamma$, skew) pair. We chose a discrete action space for our experiment to apply RL to manipulate AS-related parameters, aiming keep the algorithm as simple and quickly trainable as possible. A continuous action space, as the one used to choose spread values in [28], may possibly perform better, but the algorithm would be more complex and the training time greater.

The AS model has further parameters to set: the time interval, $\pi$, for the estimation of order book liquidity ($k$), and the window, $w$, of orderbook ticks to consider when determining market volatility (as the standard deviation of the mid-price, $\sigma$). Unlike $\gamma$ and skew, the values for $\pi$ and $w$ are not set through actions of the Alpha-AS agent. Instead, they are fixed at the values reached by genetic selection for the direct AS model (see Section 4.2).

**4.1.6 Reward ($R$).** The reward is given by the **Asymmetric dampened P&L** [23, 37] (Eq (15)). This is obtained from the algorithm's P&L, discounting the losses from speculative positions. The Asymmetric dampened P&L penalizes speculative positions, as speculative profits are not added while losses are discounted.

$$R(t_i) = \Psi(\tau_i) - max[0, I(\tau_i)\Delta m(\tau_i)] \tag{15}$$

where $\Psi(\tau_i)$ is the open P&L for the 5-second action time step, $I(\tau_i)$ is the inventory held by the agent and $\Delta m(\tau_i)$ is the speculative P&L (the difference between the open P&L and the close P&L), at time $\tau_i$, which is the end of the $i$th 5-second agent action cycle.

**4.1.7 The Alpha-AS deep Q-learning algorithm.** With the above definition of our Alpha-AS agent and its orderbook environment, states, actions and rewards, we can now revisit the reinforcement learning model introduced in Section (4.1.2) and specify the Alpha-AS RL model.

Fig 3 illustrates the model structure. The Alpha-AS agent receives an update of the orderbook (its environment) every time a market tick occurs. The Alpha-AS agent records the new market tick information by modifying the appropriate market features it keeps as part of its state representation. The agent also places one bid and one ask order in response to every tick. Once every 5 seconds, the agent records the asymmetric dampened P&L it has obtained as its reward for placing these bid and ask orders during the latest 5-second time step. Based on the market state and the agent's private indicators (i.e., its latest inventory levels and rewards), a prediction neural network outputs an action to take. As defined above, this action consists in setting the value of the risk aversion parameter, $\gamma$, in the Avellaneda-Stoikov formula to calculate the bid and ask prices, and the skew to be applied to these. The agent will place orders at the resulting skewed bid and ask prices, once every market tick during the next 5-second time step.

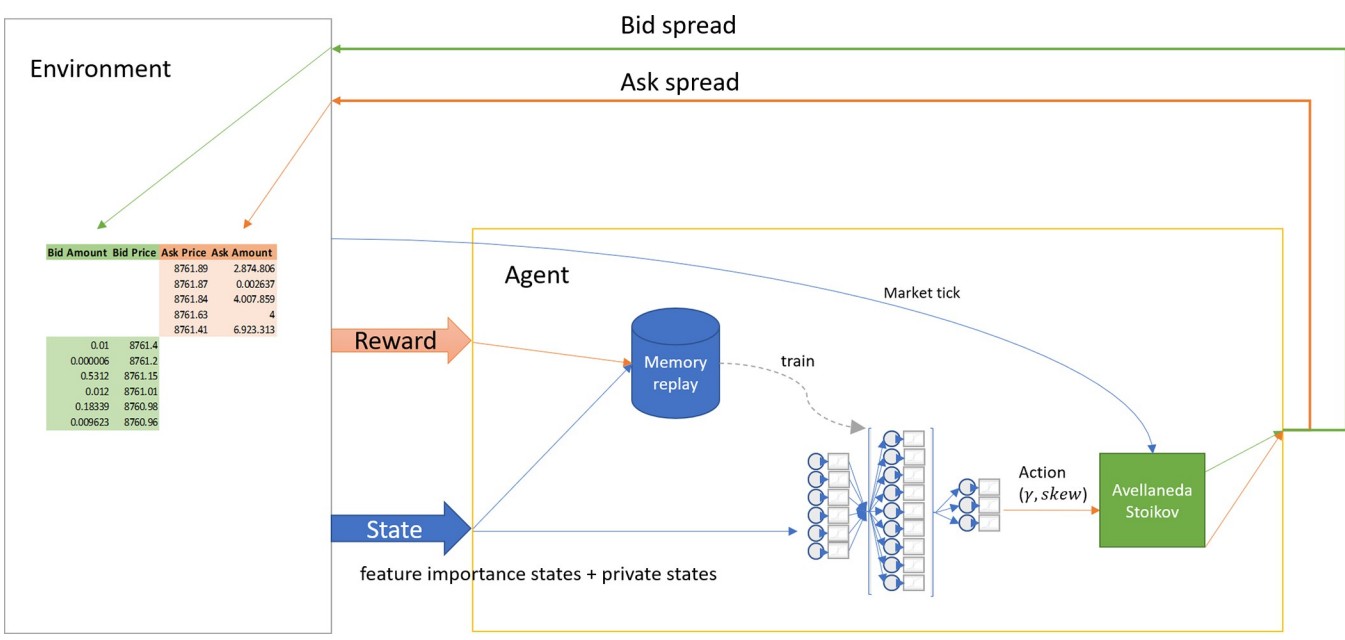

**Fig 3. The Alpha-Avellaneda-Stoikov workflow.**

Consequently, the Alpha-AS agent adapts its bid and ask order prices dynamically, reacting closely (at 5-second steps) to the changing market. This 5-second interval allows the Alpha-AS algorithm to acquire experience trading with a certain bid and ask price repeatedly under quasi-current market conditions. As we shall see in Section 4.2, the parameters for the direct Avellaneda-Stoikov model to which we compare the Alpha-AS model are fixed (using a genetic algorithm) at a parameter tuning step once every 5 days of trading data.

The reinforcement learning algorithm works as follows:

```
Initialize Q(s, a) to 0
For each episode:
    For t = 0...T:
        Record the current state, s
        Every 5 seconds (i.e., if t% Timestep = 0):
            Apply policy function:
                Choose the action, a to take from the current state, s,
using either:
                    exploration (with prob. ε): set a random (γ,skew)
pair
                    or
                    exploitation (with prob. 1−ε): obtain a (γ,skew) pair
from the neural network
            Take action a: apply the Avellaneda-Stoikov formulas with
(γ,skew)
            Update the memory replay buffer
```
$$Q_i(s,a) = (1 - \alpha)PredictionDQN(s,a) + \alpha[R(s,a) + \gamma_d(max_{a'}(TargetDQN(s',a')))]$$
```
        If t % training_predict_period = 0: train Prediction DQN
If t % training_target_period = 0: train Target DQN
```

The memory replay buffer is a 10,000×84 matrix with a column for each available action and for each of the features that describe the market states. Its rows fill up with successive experiences recorded at every market tick. Each row contains the private and market feature values defining the MDP's state, *s*; the latest rewards, *r*, associated with each of the 20 actions, when

they were last taken from that state; and the feature values defining the next state, $s'$, arrived at from $s$ by taking action $a$. When the memory replay buffer is full, the ten thousand experiences recorded in it are used to train the prediction DQN. Subsequently, this network is trained with a new batch of experiences every 4 hours (in trading data time). The target DQN is trained from the prediction DQN (the former is a copy of the latter) once every 2 training steps of the prediction network (i.e., every 8 hours-worth of trading data).

At the start of every 5-second time step, the latest state (as defined in Section 4.1.4) is fed as input to the prediction DQN. The sought-after Q values–those corresponding to past experiences of taking actions from this state– are then computed for each of the 20 available actions, using both the prediction DQN and the target DQN (Eq (9)).

An $\varepsilon$-greedy policy is followed to determine the action to take during the next 5-second window, choosing between exploration (random action selection), with probability $\varepsilon$, and exploitation (selection of the action currently with the highest Q value), with probability 1-$\varepsilon$. The selected action is then taken repeatedly, once every market tick, in the following 5-second window, at the end of which the reward (the Asymmetric Dampened P&L) obtained from this repeated execution of the action is computed.

*Neural network architectures*. The prediction DQN receives as input the state-defining features, with their values normalised, and it outputs a value between 0 and 1 for each action. Hence, it has 32 input neurons (one per feature) and 20 output neurons (one per action available to the agent). The DQN has two hidden layers, each with 104 neurons, all applying a ReLu activation function. The output layer neurons perform linear activation.

At each training step (every 4 hours) the parameters of the prediction DQN are updated using gradient descent. An early stopping strategy is followed on 25% of the training sets to avoid overfitting. The architecture of the target DQN is identical to that of the prediction DQN, the parameters of the former being copied from the latter every 8 hours.

We tested two variants of our Alpha-AS model, differing in the architecture of their hidden layers. Initial tests with a DNN featuring two dense hidden layers were followed by tests using a RNN with two LSTM (long short-term memory) hidden layers, encouraged by results reported using this architecture [26, 38].

## 4.2 Gen-AS: Avellaneda-Stoikov model with genetically tuned parameters

There are two basic parameters to be determined in our direct Avellaneda-Stoikov model (Eqs (1)–(3)): risk aversion ($\gamma$) and the time interval, $\pi$, for the estimation of the order book liquidity parameter ($k$) (no further quantities need to be specified in our AS model, as discussed in Section 2). We also need to specify the size of the window of ticks, $w$, used to estimate volatility ($\sigma$). The size of this parameter space is large, and we need to find the values that make the AS model perform as close to optimally as possible. One way to achieve this would be to calibrate the parameters using closed formulas derived from reasonable statistical models, in the line explored by Fernández-Tapia [4] Another option is to rely on genetic algorithms, which have been applied widely to calibrate machine learning models [39–41]. In algorithmic trading they are commonly used to find the parameter values of a trading model that produce the highest profit [42]. This motivated us to lean on a genetic algorithm to find the best-performing values for our parameters [43].

The genetic algorithm described below decides the values for the parameters throughout the test period, based on the relative performance over the latest full day of trading achieved by a population of models with differing values for their parameters To underline the biological metaphor, the set of parameters, or *genes*, on which the model is being tuned is called a *chromosome*. Genetic algorithms compare the performance of a population of copies of a model,

each with random variations, called *mutations*, in the values of the genes present in its chromosomes. The best-performing models, that is, the model instances which achieve the highest score on a *fitness function*, are selected to create from them a new generation of models by introducing further mutations and by mixing the chromosomes of the selected parent models, a procedure referred to as *crossover*. This process of random mutation, crossover, and selection of the fittest is iterated over a number of generations, with the genetic pool gradually evolving. Finally, the best-performing model overall, with its corresponding parameter values contained in its chromosome, is retained for subsequent application to the problem at hand. In our case, it will be the AS model used as a baseline against which to compare the performance of our Alpha-AS model.

**Parameters and data.** For our Gen-AS model we define a chromosome with three genes, corresponding to the aforementioned parameters. We seek the best-performing values for these parameters, within the following ranges (in which we deem the values are reasonable):

- Risk aversion ($\gamma$): [0.01, 0.9].

- Time interval ($\pi$) to estimate $k$: [1, 10].

- Tick window size ($w$): [5, 25].

Our fitness function is the Sharpe ratio, defined as follows:

$$Sharpe\ ratio = mean(returns)/std(returns) \qquad (16)$$

We performed genetic search at the beginning of the experiment, aiming to obtain the values of the AS model parameters that yield the highest Sharpe ratio, working on the same orderbook data.

**Procedure.** Our algorithm works through 10 generations of instances of the AS model, which we will refer to as *individuals*, each with a different chromosomal makeup (parameter values). In the first generation, 45 individuals were created by assigning to each of the four genes random values within the defined ranges. These individuals run (in parallel) through the orderbook data, and are then ranked according to the Sharpe ratio they have attained. For each subsequent generation 45 new individuals run through the data and then added to the cumulative population, retaining all the individuals from previous generations. The 10 generations thus yield a total of 450 individuals, ranked by their Sharpe ratio. Note that, since we retain all individuals from generation to generation, the highest Sharpe ratio the cumulative population never decreases in subsequent generations.

The chromosomes of the 45 individuals that are added to the pool in each generation are determined as follows. An average of 70% of chromosomes are created by crossover and 30% by mutation. More precisely, each new chromosome has a probability of 0.7 of being created by crossover and of 0.3 of being created by mutation. We now describe how our mutation and crossover mechanisms work:

*Mutation (asexual reproduction).* A single parent individual is selected randomly from the current population (all the individuals created so far in previous generations), with a selection probability proportional to the Sharpe score it has achieved (thus, higher-scoring individuals have a greater probability of passing on their genes). The chromosome of the selected individual is then extracted and a truncated Gaussian noise is applied to its genes (truncated, so that the resulting values don't fall outside the defined intervals). The new genetic values form the chromosome of the offspring model. The mean of the added Gaussian noise is 0; its standard deviation starts at twice the value range of each of the genes, to generate the second-generation offspring, and it is decreased exponentially to generate each subsequent generation (the standard deviation is one hundredth of the gene's value range, to generate the 10th generation).

*Crossover (sexual reproduction)*. Two parents are selected from the current population. Again, the probability of selecting a specific individual for parenthood is proportional to the Sharpe ratio it has achieved. A weighted average of the values of the two parents' genes is then computed.

Let $\gamma_x$, $w_x$ *and* $k_x$ be the parameter values of the first parent, $x$, and $\gamma_y$, $w_y$ *and* $k_y$ the genes of the second parent, $y$. The genes of the offspring, $O$, will be determined as:

$$\gamma_O = a\gamma_x + (1 - a)\gamma_y \tag{17}$$

$$w_O = bw_x + (1 - b)w_y \tag{18}$$

$$\pi_O = c\pi_x + (1 - c)\pi_y \tag{19}$$

where $a$, $b$, $c$ and $d$ are random values between 0.2 and 0.8.

**Initial parameter tuning results.** The data for the first use of the genetic algorithm was the full day of trading on 8th December 2020.

The parameter values of this best-performing instance of the AS model are the following:

- Risk aversion: $\gamma = 0.624$.

- Tick window size: $w = 25$.

- Time interval: $\pi = 1$ minute.

- As stated in Section 4.1.7, these values for $w$ and $k$ are taken as the fixed parameter values for the Alpha-AS models. They are not recalibrated periodically for the Gen-AS so that their values do not differ from those used throughout the experiment in the Alpha-AS models. If $w$ and $k$ were different for Gen-AS and Alpha-AS, it would be hard to discern whether observed differences in the performance of the models are due to the action modifications learnt by the RL algorithm or simply the result of differing parameter optimisation values. Alternatively, $w$ and $k$ could be recalibrated periodically for the Gen-AS model and the new values introduced into the Alpha-AS models as well. However, this would require discarding the prior training of the latter every time $w$ and $k$ are updated, forcing the Alpha-AS models to restart their learning process every time.

## 5. Experimental setup

All tests and training were run on the same computer, with an *AMD Ryzen Threadripper 2990WX 3.0GHz* CPU and 64GB of RAM, running on windows 10 x64 with python 3.6 and java 1.8.

### 5.1 Data and test procedure

The dataset used contains the L2 orderbook updates and market trades from the btc-usd (bitcoin–dollar pair), for the period from 7th December 2020 to 8th January 2021, with 12 hours of trading data recorded for each day. Most of the data, the Java source code and the results are accessible from the project's GitHub repository [44].

For every day of data the number of ticks occurring in each 5-second interval had positively skewed, long-tailed distributions. The means of these thirty-two distributions ranged from 33 to 110 ticks per 5-second interval, the standard deviations from 21 to 67, the minimums ran from 0 to 20, the maximums from 233 to 1338, and the skew ranged from 1.0 to 4.4.

The btc-usd data for 7th December 2020 was used to obtain the feature importance values with the MDI, MDA and SFI metrics, to select the most important features to use as input to the Alpha-AS neural network model.

The btc-usd data for the following day, 8th December 2020, was used for two purposes:

- To start filling Alpha-AS memory replay buffer and training the model (Section 5.2).

- To perform the first genetic tuning of the baseline AS model parameters (Section 4.2).

The resulting Gen-AS model, two non-AS baselines (based on Gašperov [25]) and the two Alpha-AS model variants were run with the rest of the dataset, from 9th December 2020 to 8th January 2021 (30 days), and their performance compared.

## 5.2 Training

In the training phase we fit our two Alpha-AS models with data from a full day of trading (8th December 2020). In this, the most time-consuming step of the backtest process, our algorithms learned from their trading environment what AS model parameter values to choose every five seconds of trading (in those 5 seconds; see Section 4.1.3).

We were able to achieve some parallelisation by running five backtests simultaneously on different CPU cores. Each process filled its own memory replay buffer. Upon finalization of the five parallel backtests, the five respective memory replay buffers were merged. This constituted one training iteration. 10 such training iterations were completed, all on data from the same full day of trading, with the memory replay buffer resulting from each iteration fed into the next. The replay buffer obtained from the final iteration was used as the initial one for the test phase. At this point the trained neural network model had 10,000 rows of experiences and was ready to be tested out-of-sample against the baseline AS models.

The training time for each Alpha-AS model was approximately 7 hours.

## 5.3 Test models and performance indicators

We compared the performance of our two Alpha-AS model variants with three baseline models. To reiterate, our two Alpha-AS double DQN architectures differed as follows:

- Alpha-AS-1 uses a DNN with two dense hidden layers.

- Alpha-AS-2 uses a RNN with two LSTM hidden layers.

  Our three baseline models:

- AS-Gen: the Avellaneda-Stoikov model with genetically tuned parameters (described in Section 4.2).

- Fixed Offset with Inventory Constraints (Constant Spread) [25]: FOIC is a constant spread model that places a buy order and a sell order on the first level of the orderbook, until an inventory limit is reached. When this happens, only one side of the algorithm operates (buy or sell), in order to offset the inventory and so reduce market risk.

- Linear in Inventory with Inventory Constraints (Linearly constant spread) [25]: LIIC is also a constant linear spread algorithm that can place first-level quotes on both sides of the market. It differs from FOIC in its inventory offset strategy to reduce risk: in LIIC the quantity of the buy (sell) orders is decreased linearly as the positive (negative) inventory increases. When a positive (negative) inventory threshold is reached, buy (sell) orders are interrupted.

The following performance indicators were used to compare the models at the end of each test day:

- Sharpe ratio: a measure of risk-adjusted return (given by Eq (16)). The Sharpe ratio contrasts returns against their volatility, penalizing higher values of the latter (regardless of whether the returns are positive or negative).

- Sortino ratio: a variation of the Sharpe ratio that penalizes the volatility of negative returns only (Eq (20)).

$$Sortino\ ratio = \frac{mean(returns)}{std(negative\ returns)} \tag{20}$$

- Maximum drawdown (Max DD) [25]: the largest drop in portfolio value between any two instants throughout the current test day (less is better).

- P&L to Mean Absolute Position ratio (P&L-to-MAP) [25]: a measure of return (the Open P&L) relative to inventory size, $I$ (Eq (16)). Lower inventory levels, whether positive or negative, yield higher P&L-to-MAP values, reflecting the lower risk.

$$P\&L\ to\ Map = \frac{\Psi(t_i)}{mean(|I|)} \tag{21}$$

## 6 Results

The performance results for the 30 days of testing of the two Alpha-AS models against the three baseline models are shown in Tables 2–5. All ratios are computed from Close P&L returns (Section 4.1.6), except P&L-to-MAP, for which the open P&L is used. Figures in bold (underlined) are the best (second best) values among the five models for the corresponding test days. Figures for Alpha-AS 1 and 2 are given in green (red) if their value is higher (lower) than that for the AS-Gen model for the same day. Higher (green) is better than lower (red) for the Sharpe ratio, the Sortino ratio and P&L-to-MAP, while the opposite is true for Max DD. The bottom row ('Days best') in each table totals the number of days for which each model achieved the best score for the corresponding performance indicator. Figures in parenthesis are the number of days the Alpha-AS model in question (1 or 2) was second best only to the other Alpha-AS model (and therefore would have computed another overall 'win' had it competed alone against the baseline and AS-Gen models).

Tables 2 to 5 show performance results over 30 days of test data, by indicator (**2.** Sharpe ratio; **3.** Sortino ratio; **4.** Max DD; **5.** P&L-to-MAP), for the two baseline models (FOIC and LIIC), the Avellaneda-Stoikov model with genetically optimised parameters (AS-Gen) and the two Alpha-AS models.

Table 6 compares the results of the Alpha-AS models, combined, against the two baseline models and Gen-AS. The figures represent the percentage of wins of one among the models in each group against all the models in the other group, for the corresponding performance indicator.

A Kruskal-Wallis test shows that there are strongly significant differences across the models for each of the four daily performance indicators ($H(4)_{Sharpe} = 66.22$, $H(4)_{Sortino} = 66.10$, $H(4)_{Max-DD} = 54.80$, $H(4)_P\&L - to - MAP = 106.30$; $p < 10^{-10}$ in all cases).

**Table 2. Sharpe ratio.**

| Day | FOIC | LIIC | AS-Gen | Alpha-AS 1 | Alpha-AS 2 |
|---|---|---|---|---|---|
| 1 | -0.36 | -0.31 | -0.39 | -0.10 | **0.10** |
| 2 | -0.53 | -0.51 | -0.38 | **-0.05** | -0.22 |
| 3 | -0.29 | -0.32 | -0.25 | -0.18 | **-0.02** |
| 4 | -0.27 | -0.36 | -0.32 | **0.04** | -0.10 |
| 5 | -0.47 | -0.51 | -0.43 | **-0.06** | -0.07 |
| 6 | -0.53 | -0.52 | -0.42 | -0.31 | **0.02** |
| 7 | -0.42 | -0.60 | -0.51 | -0.17 | **-0.07** |
| 8 | -0.39 | -0.46 | -0.28 | -0.18 | **-0.08** |
| 9 | -0.29 | -0.52 | -0.39 | **-0.01** | -0.09 |
| 10 | -0.57 | -0.51 | -0.27 | -0.46 | **-0.23** |
| 11 | -0.32 | -0.38 | -0.43 | -0.16 | **-0.06** |
| 12 | -0.27 | -0.57 | -0.24 | -0.07 | **0.01** |
| 13 | -0.43 | -0.32 | -0.24 | **-0.01** | -0.15 |
| 14 | -0.51 | -0.30 | -0.20 | **0.17** | -0.29 |
| 15 | -0.37 | -0.29 | -0.26 | -0.26 | **-0.12** |
| 16 | -0.54 | -0.14 | -0.41 | **0.18** | -0.06 |
| 17 | -0.51 | -0.40 | -0.13 | -0.11 | **-0.03** |
| 18 | -0.46 | -0.27 | -0.22 | **-0.04** | -0.12 |
| 19 | -0.51 | -0.47 | -0.21 | -0.07 | **0.00** |
| 20 | -0.30 | -0.31 | -0.06 | **-0.03** | -0.04 |
| 21 | -0.57 | -0.44 | **0.10** | -0.41 | -0.03 |
| 22 | -0.49 | **0.02** | -0.21 | -0.12 | -0.19 |
| 23 | -0.57 | -0.52 | **-0.28** | -0.32 | **-0.28** |
| 24 | -0.42 | -0.50 | **-0.31** | -0.36 | -0.48 |
| 25 | -0.51 | -0.30 | -0.04 | **0.19** | -0.01 |
| 26 | -0.51 | -0.41 | -0.04 | -0.02 | **0.06** |
| 27 | -0.35 | -0.06 | -0.01 | **0.00** | -0.38 |
| 28 | -0.30 | **-0.08** | -0.18 | -0.29 | -0.28 |
| 29 | -0.56 | -0.16 | **0.04** | -0.15 | -0.20 |
| 30 | -0.31 | -0.39 | -0.27 | **-0.04** | -0.43 |
| **Days best** | **0** | **2** | **4** | **12** (+11) | **12** (+7) |
| **Median** | -0.45 | -0.39 | -0.26 | -0.09 | -0.09 |
| **Mean** | -0.43 | -0.36 | -0.24 | -0.11 | -0.13 |
| **Std. Dev.** | 0.10 | 0.16 | 0.15 | 0.16 | 0.14 |

Post-hoc Mann-Whitney tests were conducted to analyse selected pairwise differences between the models regarding these performance indicators. The results are summarised in Table 7.

## Sharpe ratio

The Sharpe ratio is a measure of mean returns that penalises their volatility. Table 2 shows that one or the other of the two Alpha-AS models achieved better Sharpe ratios, that is, better risk-adjusted returns, than all three baseline models on 24 (12+12) of the 30 test days. Furthermore, on 9 of the 12 days for which Alpha-AS-1 had the best Sharpe ratio, Alpha-AS-2 had the second best; conversely, there are 11 instances of Alpha-AS-1 performing second best after

**Table 3. Sortino ratio.**

| Day | FOIC | LIIC | AS-Gen | Alpha-AS 1 | Alpha-AS 2 |
|---|---|---|---|---|---|
| 1 | -0.34 | -0.30 | -0.38 | -0.10 | **0.23** |
| 2 | -0.47 | -0.46 | -0.36 | **-0.08** | -0.22 |
| 3 | -0.29 | -0.32 | -0.26 | -0.18 | **-0.02** |
| 4 | -0.27 | -0.35 | -0.34 | **0.10** | -0.10 |
| 5 | -0.44 | -0.46 | -0.42 | **-0.06** | -0.07 |
| 6 | -0.48 | -0.47 | -0.40 | -0.30 | **0.03** |
| 7 | -0.39 | -0.52 | -0.46 | -0.16 | **-0.07** |
| 8 | -0.37 | -0.42 | -0.27 | -0.19 | **-0.08** |
| 9 | -0.28 | -0.48 | -0.38 | **-0.02** | -0.09 |
| 10 | -0.50 | -0.47 | -0.28 | -0.42 | **-0.23** |
| 11 | -0.31 | -0.36 | -0.39 | -0.16 | **-0.07** |
| 12 | -0.26 | -0.50 | -0.24 | -0.08 | **0.01** |
| 13 | -0.40 | -0.32 | -0.25 | **-0.01** | -0.21 |
| 14 | -0.46 | -0.29 | -0.20 | **0.47** | -0.28 |
| 15 | -0.35 | -0.28 | -0.27 | -0.27 | **-0.12** |
| 16 | -0.49 | -0.16 | -0.41 | **1.06** | -0.06 |
| 17 | -0.45 | -0.38 | -0.14 | -0.15 | **-0.03** |
| 18 | -0.43 | -0.26 | -0.23 | **-0.07** | -0.12 |
| 19 | -0.47 | -0.44 | -0.24 | -0.07 | **0.00** |
| 20 | -0.32 | -0.31 | -0.08 | **-0.04** | -0.05 |
| 21 | -0.50 | -0.41 | **0.19** | -0.39 | -0.03 |
| 22 | -0.45 | **0.04** | -0.25 | -0.12 | -0.20 |
| 23 | -0.50 | -0.47 | -0.29 | -0.32 | **-0.27** |
| 24 | -0.39 | -0.46 | **-0.30** | -0.34 | -0.43 |
| 25 | -0.46 | -0.29 | -0.04 | **0.46** | -0.01 |
| 26 | -0.47 | -0.39 | -0.05 | -0.05 | **0.08** |
| 27 | -0.34 | -0.08 | -0.01 | **0.00** | -0.36 |
| 28 | -0.29 | **-0.11** | -0.21 | -0.30 | -0.28 |
| 29 | -0.49 | -0.16 | **0.06** | -0.15 | -0.19 |
| 30 | -0.30 | -0.37 | -0.28 | **-0.08** | -0.40 |
| **Days best** | **0** | **2** | **3** | **12 (+10)** | **13 (+9)** |
| **Median** | -0.42 | -0.37 | -0.27 | -0.09 | -0.09 |
| **Mean** | -0.40 | -0.34 | -0.24 | -0.07 | -0.12 |
| **Std. Dev.** | 0.08 | 0.14 | 0.15 | 0.29 | 0.15 |

Alpha-AS-2. Thus, the Alpha-AS models came 1st *and* 2nd on 20 out of the 30 test days (67%). The AS-Gen model was a distant third, with 4 wins on Sharpe. The mean and the median of the Sharpe ratio over all test days was better for both Alpha-AS models than for the Gen-AS model (although the statistical significance of the difference was at best marginal after Bonferroni correction), and in turn the Gen-AS model performed significantly better on Sharpe than the two non-AS baselines.

## Sortino ratio

Similarly, on the Sortino ratio, one or the other of the two Alpha-AS models performed better, that is, obtained better negative risk-adjusted returns, than all the baseline models on 25 (12 +13) of the 30 days. Again, on 9 of the 12 days for which Alpha-AS-1 had the best Sharpe ratio,

**Table 4. Maximum drawdown.**

| Day | FOIC | LIIC | AS-Gen | Alpha-AS 1 | Alpha-AS 2 |
|---|---|---|---|---|---|
| 1 | 39.41 | 39.56 | 4.05 | **0.07** | 6.99 |
| 2 | 28.72 | 31.20 | 3.87 | **0.64** | 7.66 |
| 3 | 31.36 | 10.47 | 0.45 | 2.61 | **0.00** |
| 4 | 18.29 | 20.47 | 3.05 | **0.08** | 8.09 |
| 5 | 27.76 | 35.76 | 4.04 | **0.10** | 0.16 |
| 6 | 17.37 | 16.20 | 2.97 | 5.07 | **0.71** |
| 7 | 17.59 | 25.05 | 6.68 | 0.39 | **0.11** |
| 8 | 96.81 | 90.21 | 22.48 | 4.97 | **0.23** |
| 9 | 111.75 | 162.30 | 6.48 | **0.06** | 127.32 |
| 10 | 94.41 | 77.03 | **1.11** | 45.51 | 9.76 |
| 11 | 95.33 | 149.33 | 18.17 | 4.64 | **0.28** |
| 12 | 24.03 | 60.17 | 4.57 | **2.13** | 5.70 |
| 13 | 69.68 | 30.26 | **3.39** | 1,869.89 | 12.07 |
| 14 | 92.46 | 43.99 | **3.35** | 3.74 | 59.63 |
| 15 | 43.85 | 24.67 | 2.88 | 41.24 | **2.42** |
| 16 | 38.43 | 5.49 | 3.74 | **0.22** | 0.48 |
| 17 | 131.80 | 101.02 | 1.62 | 16.98 | **0.03** |
| 18 | 141.45 | 56.23 | 9.43 | 6.48 | **1.16** |
| 19 | 200.47 | 259.65 | **3.94** | 118.57 | 21.22 |
| 20 | 21.76 | 22.28 | 0.86 | **0.03** | 0.13 |
| 21 | 93.37 | 61.43 | 1.98 | 45.03 | **0.01** |
| 22 | 118.91 | **1.02** | 4.45 | 1.15 | 28.48 |
| 23 | 64.97 | 68.67 | **4.24** | 21.86 | 32.02 |
| 24 | 476.14 | 703.11 | **26.64** | 153.98 | 509.06 |
| 25 | 222.26 | 115.24 | **0.03** | 9.48 | 20.24 |
| 26 | 555.30 | 245.81 | **0.84** | 65.28 | 97.17 |
| 27 | 200.44 | 13.29 | **0.19** | 6.37 | 110.83 |
| 28 | 84.33 | 6.23 | **3.90** | 42.85 | 67.48 |
| 29 | 353.84 | 27.23 | 3.27 | **1.94** | 455.20 |
| 30 | 309.28 | 365.19 | **14.28** | 108.75 | 554.01 |
| **Days best** | **0** | **1** | **11** | **9** (+4) | **9** (+3) |
| **Median** | 92.92 | 41.78 | 3.81 | 5.02 | 7.88 |
| **Mean** | 127.39 | 95.62 | 5.57 | 86.00 | 71.29 |
| **Std. Dev.** | 135.60 | 142.99 | 6.48 | 339.22 | 152.00 |

Alpha-AS-2 had the second best; and for 10 of the 13 test days for which after Alpha-AS-2 obtained the best Sortino ratio, Alpha-AS-1 performed second best. *Both* Alpha-AS models performed better than the rest on 19 days. Meanwhile, AS-Gen, again the best of the rest, won on Sortino on only 3 test days. The mean and the median of the Sortino ratio were better for both Alpha-AS models than for the Gen-AS model (again with only marginal statistical significance), and for the latter it was significantly better than for the two non-AS baselines.

## Maximum drawdown

Maximum drawdown (Max DD) registers the largest loss of portfolio value registered between any two points of a full day of trading. By identifying the largest losses from any peak within

**Table 5. P&L-to-MAP.**

| Day | FOIC | LIIC | AS-Gen | Alpha-AS 1 | Alpha-AS 2 |
|---|---|---|---|---|---|
| 1 | -201,046.82 | -124,815.40 | -1,643.93 | -0.20 | 21.51 |
| 2 | -37,830.20 | -244,586.13 | -1,064.75 | -1.10 | -16.40 |
| 3 | -85,397.04 | -10,492.71 | -13.44 | -5.96 | -0.01 |
| 4 | -76,534.03 | -39,925.66 | -3,370.04 | 0.20 | -18.10 |
| 5 | -31,910.31 | -91,931.80 | -1,171.24 | -0.15 | -0.30 |
| 6 | -73,352.04 | -21,918.20 | -1,053.34 | -14.44 | 7.50 |
| 7 | -29,077.48 | -87,900.74 | -39,443.20 | -0.65 | -0.18 |
| 8 | -114,697.71 | -163,935.67 | -43,171.85 | -11.53 | -0.36 |
| 9 | -265,431.69 | -2,658,430.60 | -47,010.74 | -0.04 | -300.74 |
| 10 | -679,194.37 | -666,451.16 | -37.22 | -283.04 | -19.79 |
| 11 | -180,195.11 | -316,767.04 | -15,074.68 | -7.21 | -0.62 |
| 12 | -153,020.33 | -204,990.67 | -50,665.89 | -2.37 | 2.46 |
| 13 | -280,686.62 | -92,545.21 | -5,358.47 | -14,041.08 | -54.67 |
| 14 | -348,744.49 | -132,878.74 | -2,080.53 | 62.85 | -121.49 |
| 15 | -238,994.79 | -62,593.42 | -1,221.36 | -1,001.73 | -3.29 |
| 16 | -85,973.96 | -6,453.16 | -2,522.39 | 10.60 | -0.70 |
| 17 | -583,164.73 | -2,118,249.56 | -2,779.62 | -26.71 | -0.04 |
| 18 | -255,274.76 | -262,997.46 | -16,438.08 | -7.40 | -1.70 |
| 19 | -973,167.02 | -539,787.43 | -2,288.63 | -590.29 | -1.57 |
| 20 | -18,379.71 | -40,276.07 | -1,939.48 | -0.06 | -0.19 |
| 21 | -258,629.43 | -133,033.02 | 1,695.84 | -287.66 | -0.01 |
| 22 | -311,208.74 | 749.20 | -5,558.30 | -2.83 | -44.51 |
| 23 | -62,458.91 | -176,342.18 | -9,187.70 | -153.45 | -51.84 |
| 24 | -59,514,101.49 | -1,377,661.44 | -93,473.30 | -1,490.48 | -1,531.04 |
| 25 | -163,066.66 | -693,103.94 | -0.88 | 163.63 | -3.98 |
| 26 | -9,915,984.62 | -758,394.51 | -414.18 | -67.99 | 298.49 |
| 27 | -171,010.82 | -9,592.09 | -0.97 | 0.41 | -280.94 |
| 28 | -69,672.79 | -9,169.87 | -12,425.81 | -801.50 | -152.62 |
| 29 | -236,205.98 | -49,300.73 | 2,309.20 | -5.30 | -1,282.23 |
| 30 | -173,457.54 | -746,529.51 | -13,023.30 | -136.72 | -1,404.35 |
| **Days best** | **0** | **1** | **2** | **11** (+14) | **16** (+9) |
| **Median** | -176,826.33 | -132,955.88 | -2,405.51 | -6.59 | -2.50 |
| **Mean** | -2,519,595.67 | -394,676.83 | -12,280.94 | -623.41 | -165.39 |
| **Std. Dev.** | 10,910,922.98 | 630,895.76 | 21,519.58 | 2,559.30 | 433.33 |

each day, this indicator can be leveraged to monitor and learn from downward trends in rewards that are longer stretching than those captured by the Sortino ratio, and penalize the actions that led to them in the market context in which they were taken.

On this performance indicator, AS-Gen was the overall best performing model, winning on 11 days. The mean Max DD for the AS-Gen model over the entire test period was visibly the

**Table 6. Number of days either Alpha-AS-1 or Alpha-AS-2 scored best out of all tested models, for each of the four performance indicators.**

| | Sharpe | Sortino | Max DD | P&L Map |
|---|---|---|---|---|
| 1st and 2nd place days for Alpha-AS 1 & 2 | 20 | 19 | 7 | 23 |

**Table 7. Mann-Whitney tests comparing the four daily performance indicator values (Sharpe, Sortino, Max DD and P&L-to-MAP) obtained for the Gen-AS model with the corresponding values obtained for the other models, over the 30 test days.** (Reported: Mann-Whitney U, significance level (p, with Bonferroni correction) and effect size ($r = Z/\sqrt{30}$)).

| Comparison | Performance indicator | | | |
|---|---|---|---|---|
| Gen-AS vs.: | Sharpe | Sortino | Max DD | P&L-to-MAP |
| FOIC | $U = 128.5, p < 10^{-5}, r = 0.87$ | $U = 139.5, p < 10^{-4}, r = 0.84$ | $U = 11, p < 10^{-9}, r = 1.18$ | $U = 26, p < 10^{-8}, r = 1.14$ |
| LIIC | $U = 238, p < .05, r = 0.57$ | $U = 247, p < .05, r = 0.55$ | $U = 56, p < 10^{-7}, r = 1.06$ | $U = 85, p < 10^{-6}, r = 0.96$ |
| Alpha-AS-1 | $U = 253, p < .1, r = -0.53$ | $U = 255.5, p < .1, r = -0.53$ | $U = 378.5, p > .2$ | $U = 147, p < 10^{-4}, r = -0.82$ |
| Alpha-AS-2 | $U = 260.5, p < .1, r = -0.51$ | $U = 244, p < .05, r = -0.56$ | $U = 366.5, p > .2$ | $U = 85, p < 10^{-4}, r = -0.86$ |

lowest (best), and its standard deviation was also the lowest by far from among all models. In comparison, both the mean and the standard deviation of the Max DD for the Alpha-AS models were very high. We note that the fact that the standard deviation was so high for the Alpha-AS models, and accounting for the day victories Alpha-AS 1 and 2 'stole' from one another, they would have achieved the best Max DD performance for 13 and 12 of the test days, respectively, both slightly better than AS-Gen. Indeed, the differences in Max DD performance between Gen-AS and either of the Alpha-AS models, over all test days, are not statistically significant, despite the large differences in means. The latter are a result of extreme outliers for the Alpha-AS models from days in which these obtained a very poor (i.e., high) value for Max DD. The medians, however, are very similar to the median for the Gen-AS model.

Nevertheless, it is still interesting to note that AS-Gen performs much better on this indicator than on the others, relative to the Alpha-AS models. To understand why this may be so, we recall that AS-Gen does not alter the risk aversion parameter after it has been set through genetic selection to the value that performs best on the initial test data, nor does it modify the spread given by the AS formulas, which is mathematically optimal to the extent that its parameter values are realistic. This means that, *provided its parameter values describe the market environment closely enough*, the pure AS model is guaranteed to output the bid and ask prices that minimise inventory risk, and *any* deviation from this strategy will entail a greater risk. Throughout a full day of trading, it is more likely than within shorter time frames that there will be intervals at which the market is indeed closely matched by the AS formula parameters. The greater inventory risk taken by the Alpha-AS models during such intervals can be punished with greater losses. Occasionally the losses may be large (as an example, Table 4 reveals that Alpha-AS-1 suffered an exceptionally large Max DD of 1,869.89 on test day 13), though further testing would be required to ascertain whether or not these extreme values are actually outliers due to chance alone. Conversely, the gains may also be greater, a benefit which is indeed reflected unequivocally in the results obtained for the P&L-to-MAP performance indicator.

## P&L-to-MAP

On the P&L-to-MAP ratio, Alpha-AS-1 was the best-performing model for 11 test days, with Alpha-AS-2 coming second on 9 of them, whereas Alpha-AS-2 was the best-performing model on P&L-to-MAP for 16 of the test days, with Alpha-AS-1 coming second on 14 of these. Here the single best-performing model was Alpha-AS-2, winning for 16 days and coming second on 10 (on 9 of which losing to Alpha-AS-1). Alpha-AS-1 had 11 victories and placed second 16 times (losing to Alpha-AS-2 on 14 of these). AS-Gen had the best P&L-to-MAP ratio only for 2 of the test days, coming second on another 4. The mean and the median P&L-to-MAP ratio were very significantly better for both Alpha-AS models than the Gen-AS model.

**Table 8. Comparison of values for Max DD and P&L-to-MAP between the Gen-AS model and the Alpha-AS models (αAS1 and αAS2).** The "Sign comparison of value differences" side of the table (right) highlights in green (Alpha-AS "better") the test days for which the respective Alpha-AS models performed worse on Max DD but better on P&L-to-MAP relative to the Gen-AS model, the latter being the more desirable indicator in which to perform well (since maximizing the P&L profile is the central point of the AS method). Conversely, test days for which the Alpha-ASs did worse than Gen-AS on P&L-to-MAP in spite of performing better on Max DD are highlighted in red (Alpha-AS "worse").

| Day | Value difference (GenAS–αASx) | | | | Sign comparison of value differences between Max DD and P&L-to-MAP | |
|---|---|---|---|---|---|---|
| | Max DD | | P&L-to-MAP | | | |
| | αAS1 | αAS2 | αAS1 | αAS2 | αAS1 | αAS2 |
| 1 | 3.98 | -2.94 | 1,643.73 | 1,665.44 | Same | Opposite (α better) |
| 2 | 3.23 | -3.79 | 1,063.65 | 1,048.35 | Same | Opposite (α better) |
| 3 | -2.16 | 0.45 | 7.48 | 13.43 | Opposite (α better) | Same |
| 4 | 2.97 | -5.04 | 3,370.24 | 3,351.94 | Same | Opposite (α better) |
| 5 | 3.94 | 3.88 | 1,171.09 | 1,170.94 | Same | Same |
| 6 | -2.10 | 2.26 | 1,038.90 | 1,060.84 | Opposite (α better) | Same |
| 7 | 6.29 | 6.57 | 39,442.55 | 39,443.02 | Same | Same |
| 8 | 17.51 | 22.25 | 43,160.32 | 43,171.49 | Same | Same |
| 9 | 6.42 | -120.84 | 47,010.70 | 46,710.00 | Same | Opposite (α better) |
| 10 | -44.40 | -8.65 | -245.82 | 17.43 | Same | Opposite (α better) |
| 11 | 13.53 | 17.89 | 15,067.47 | 15,074.06 | Same | Same |
| 12 | 2.44 | -1.13 | 50,663.52 | 50,668.35 | Same | Opposite (α better) |
| 13 | -1,866.50 | -8.68 | -8,682.61 | 5,303.80 | Same | Opposite (α better) |
| 14 | -0.39 | -56.28 | 2,143.38 | 1,959.04 | Opposite (α better) | Opposite (α better) |
| 15 | -38.36 | 0.46 | 219.63 | 1,218.07 | Opposite (α better) | Same |
| 16 | 3.52 | 3.26 | 2,532.99 | 2,521.69 | Same | Same |
| 17 | -15.36 | 1.59 | 2,752.91 | 2,779.58 | Opposite (α better) | Same |
| 18 | 2.95 | 8.27 | 16,430.68 | 16,436.38 | Same | Same |
| 19 | -114.63 | -17.28 | 1,698.34 | 2,287.06 | Opposite (α better) | Opposite (α better) |
| 20 | 0.83 | 0.73 | 1,939.42 | 1,939.29 | Same | Same |
| 21 | -43.05 | 1.97 | -1,983.50 | -1,695.85 | Same | Opposite (α worse) |
| 22 | 3.30 | -24.03 | 5,555.47 | 5,513.79 | Same | Opposite (α better) |
| 23 | -17.62 | -27.78 | 9,034.25 | 9,135.86 | Opposite (α better) | Opposite (α better) |
| 24 | -127.34 | -482.42 | 91,982.82 | 91,942.26 | Opposite (α better) | Opposite (α better) |
| 25 | -9.45 | -20.21 | 164.51 | -3.10 | Opposite (α better) | Same |
| 26 | -64.44 | -96.33 | 346.19 | 712.67 | Opposite (α better) | Opposite (α better) |
| 27 | -6.18 | -110.64 | 1.38 | -279.97 | Opposite (α better) | Same |
| 28 | -38.95 | -63.58 | 11,624.31 | 12,273.19 | Opposite (α better) | Opposite (α better) |
| 29 | 1.33 | -451.93 | -2,314.50 | -3,591.43 | Opposite (α worse) | Same |
| 30 | -94.47 | -539.73 | 12,886.58 | 11,618.95 | Opposite (α better) | Opposite (α better) |
| **Days αASx better / worse** | **14 / 16** | **12 / 18** | **26 / 4** | **26 / 4** | **13 / 1** | **15 / 1** |

On the whole, the Alpha-AS models are doing the better job at accruing gains while keeping inventory levels under control.

Table 8 provides further insight combining the results for Max DD and P&L-to-MAP. From the negative values (highlighted in red) in the Max DD columns, we see that Alpha-AS-1 had a larger Max DD (i.e., performed worse) than Gen-AS on 16 of the 30 test days. However, on 13 of those days Alpha-AS-1 achieved a better P&L-to-MAP score than Gen-AS, substantially so in many instances. Only on one day (day 29) was the trend reversed, with Gen-AS performing slightly worse than Alpha-AS-1 on Max DD, but then performing better than Alpha-AS-1 on P&L-to-MAP. The comparison with Alpha-AS-2 follows the same pattern.

From these considerations we may conclude that, while the Alpha-AS models take greater risks with their bid and ask prices, hence the comparatively poor performance on Max DD, they nevertheless obtain much better profit-to-inventory ratios (P&L-to-MAP), thus displaying superior inventory risk management compared to the baseline models.

Two important observations can be drawn from these results:

1. Gen-AS performs better than the baseline models, as expected from a model that is designed to place bid and ask prices that minimize inventory risk optimally (by mathematical construction) given a set of parameter values that are themselves optimized periodically from market data using a genetic algorithm.

2. Overall, both Alpha-AS models obtain higher and more stable returns, as well as a better P&L-to-inventory profile than AS-Gen and the non-AS baseline models. That is, they achieve a better P&L profile with less exposure to market movements.

The latter is an important feature for market maker algorithms. Indeed, this result is particularly noteworthy as the Avellaneda-Stoikov method sets as its goal precisely to minimize the inventory risk. Nevertheless, the flexibility that the Alpha-AS models are given to move and stretch the bid and ask price spread entails that the Alpha-AS models can, and sometimes do, operate locally with higher risk. This sometimes leads to poorer performance indicator values, most notably a higher Max DD. Recalling that Max DD is a high watermark record of peak-to-trough portfolio value drops throughout a full day of trading, it provides a snapshot of overall performance that reveals the Alpha-AS models may operate with more aggressive bid and ask quotes than regular AS (albeit with the non-regular feature of genetically tuned parameters). Overall performance is more meaningfully obtained from the other indicators (Sharpe, Sortino and P&L-to-MAP), which show that, at the end of the day, the Alpha-AS models' strategy pays off.

No significant differences were found between the two Alpha-AS models.

## 7 Conclusions

Reinforcement learning algorithms have been shown to be well-suited for use in high frequency trading (HFT) contexts [16, 24–26, 37, 45, 46], which require low latency in placing orders together with a dynamic logic that is able to adapt to a rapidly changing environment. In the literature, reinforcement learning approaches to market making typically employ models that act directly on the agent's order prices, without taking advantage of knowledge we may have of market behaviour or indeed findings in market-making theory. These models, therefore, must learn everything about the problem at hand, and the learning curve is steeper and slower to surmount than if relevant available knowledge were to be leveraged to guide them.

We have designed a market making agent that relies on the Avellaneda-Stoikov procedure to minimize inventory risk. It does so by acting on the risk aversion parameter of the Avellaneda-Stoikov equations and using these equations to calculate the bid and ask prices that are optimum for the chosen level of risk aversion, insofar as the other parameters in the equations reflect the market environment accurately. The agent can also skew the bid and ask prices output by the Avellaneda-Stoikov procedure, tweaking them and, by so doing, potentially counteract the limitations of a static Avellaneda-Stoikov model by reacting to local market conditions. The agent learns to adapt its risk aversion and skew its bid and ask prices under varying market behaviour through reinforcement learning using two variants (Alpha-AS-1 and Alpha-AS-2) of a double DQN architecture. The central notion is that, by relying on a procedure developed to minimise inventory risk (the Avellaneda-Stoikov procedure) by way of prior knowledge, the RL agent can learn more quickly and effectively.

A second contribution is the setting of the initial parameters of the Avellaneda-Stoikov procedure by means of a genetic algorithm working with real backtest data. This is an efficient way of arriving at quasi-optimal values for these parameters given the market environment in which the agent begins to operate. From this point, the RL agent can gradually diverge as it learns by operating in the changing market.

Backtests were performed on 30 days of bitcoin-dollar pair (BTC-USD) market data, comparing the performance of the Alpha-AS models with that of two standard baseline models and a third baseline model implementing the Avellaneda-Stoikov procedure but without a RL agent tweaking its parameters or output bid and ask prices. This Avellaneda-Stoikov baseline model (Gen-AS) constitutes another original contribution, to our knowledge, in that its parameters are optimised using a genetic algorithm working on a day's worth of data prior to the test data. The genetic algorithm selects the best-performing values (on the Sharpe ratio) found for the Gen-AS parameters on the corresponding day of data. This procedure helps establish AS parameter values that fit initial market conditions. The same set of parameters obtained for the Gen-AS model are used to specify the initial Alpha-AS models. The goal with this approach is to offer a fair comparison of the former with the latter. By training with full-day backtests on real data respecting the real-time activity latencies, the models obtained are readily adaptable for use in a real market trading environment.

The performance of the Alpha-AS models in terms of the Sharpe, Sortino and P&L-to-MAP ratios (particularly the latter) was substantially superior to that of the Gen-AS model, which in turn was superior to that of the two standard baselines. On the other hand, the performance of the Alpha-AS models on maximum drawdown varied significantly on different test days, losing to Gen-AS on over half of them, a reflection of their greater aggressiveness, made possible by their relative freedom of action. Overall, however, days of substantially better performance relative to the non-Alpha-AS models far outweigh those with poorer results, and at the end of the day the Alpha-AS models clearly achieved the best and least exposed P&L profiles. The approach, therefore, seems promising.

The results obtained suggest avenues to explore for further improvement. Drawdowns were our algorithm's most apparent weakness. It can be addressed in various ways. First, the reward function can be tweaked to penalise drawdowns more directly. Other indicators, such as the Sortino ratio, can also be used in the reward function itself. Another approach is to explore risk management policies that include discretionary rules. Alternatively, experimenting with further layers to learn such policies autonomously may ultimately yield greater benefits, as indeed may simply altering the number of layers and neurons, or the loss functions, in the current architecture.

Our motivation to continue base the trading actions on the AS formulas (rather than having the RL-based agent determine the quotes directly) is that these formulas furnish approximations to the theoretically optimal bid and ask quotes, albeit based on assumptions regarding the statistical behaviour of the market which may fall short of being realistic (as has been observed, e.g., in [25]). This potential weakness of the analytical AS approach notwithstanding, we believe the theoretical optimality of its output approximations is not to be undervalued. On the contrary, we find value in using it as a starting point from which to diverge dynamically, taking into account the most recent market behaviour.

The original Avellaneda-Stoikov model was chosen as a starting point for our research. Notable refinements of the AS approach have since been proposed, such as Guéant's [5] closed-form solution to the market maker problem for both single and multiple assets, modelling the mid-price and trades as Brownian movements, and Bergault et al.'s more recent contribution [6] also inspired by the Guéant-Lehalle-Fernandez-Tapia approximations [4]. We plan to use such approximations in further tests with our RL approach.

The training of the neural network has room for improvement through systematic optimisation of the network's parameters. Characterisation of different market conditions and specific training under them, with appropriate data (including carefully crafted synthetic data), can also broaden and improve the agent's strategic repertoire. The agent's action space itself can potentially also be enriched profitably, by adding more values for the agent to choose from and making more parameters settable by the agent, beyond the two used in the present study (i.e., risk aversion and skew). In the present study we have simply chosen the finite value sets for these two parameters that we deem reasonable for modelling trading strategies of differing levels of risk. This helps to keep the models simple and shorten the training time of the neural network in order to test the idea of combining the Avellaneda-Stoikov procedure with reinforcement learning. The results obtained in this fashion encourage us to explore refinements such as models with continuous action spaces. Similarly, the suite of state features may also be extended to include other signals, including sentiment indicators and typical HFT indicators such as Probability of Informed Trading (PIN) and Volume Synchronized Probability of Informed Trading (VPIN) that can help to uncover dynamics based on trusted trader information [35]. The logic of the Alpha-AS model might also be adapted to exploit alpha signals [47].

We relied on random forests to filter state-defining features based on their importance according to three indicators. Various techniques are worth exploring in future work for this purpose, such as PCA, Autoencoders, Shapley values [48] or Cluster Feature Importance (CFI) [49]. Other modifications to the neural network architectures presented here may prove advantageous. We mention neuroevolution to train the neural network using genetic algorithms [50] and adversarial networks [24] to improve the robustness of the market making algorithm.

In future work we will experiment combining these ideas. We also plan to compare the performance of the Alpha-AS models with that of leading RL models in the literature that do not work with the Avellaneda-Stoikov procedure.

## Author Contributions

**Conceptualization:** Javier Falces Marin, David Díaz Pardo de Vera.

**Data curation:** Javier Falces Marin.

**Formal analysis:** Javier Falces Marin, Eduardo Lopez Gonzalo.

**Funding acquisition:** Eduardo Lopez Gonzalo.

**Investigation:** Javier Falces Marin.

**Methodology:** Eduardo Lopez Gonzalo.

**Project administration:** David Díaz Pardo de Vera.

**Resources:** Javier Falces Marin.

**Software:** Javier Falces Marin.

**Supervision:** David Díaz Pardo de Vera, Eduardo Lopez Gonzalo.

**Validation:** Javier Falces Marin, Eduardo Lopez Gonzalo.

**Visualization:** Javier Falces Marin.

**Writing – original draft:** Javier Falces Marin.

**Writing – review & editing:** David Díaz Pardo de Vera, Eduardo Lopez Gonzalo.

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
