## [Decision Letter · Decision Letter 0]

23 May 2022

PONE-D-22-10287A reinforcement learning approach to improve the performance of the Avellaneda-Stoikov market-making algorithmPLOS ONE

Dear Dr. Falces Marin,

Thank you for submitting your manuscript to PLOS ONE. After careful consideration, we feel that it has merit but does not fully meet PLOS ONE’s publication criteria as it currently stands. Therefore, we invite you to submit a revised version of the manuscript that addresses the points raised during the review process.

We look forward to receiving your revised manuscript.

Kind regards,

J E. Trinidad Segovia

Section Editor

PLOS ONE

Journal Requirements:

Additional Editor Comments:

In view of the referees’ feedback and my own reading of your paper, I invite you to address all issues noted below. I consider these issues to be major in nature, requiring more than a superficial or minor revision. In particular, there are important deficiencies in the methodological section that seriously hinder the understanding of the work as well as the results obtained. Their robustness is also unclear, so I have doubts as to whether the conclusions are supported by the results presented.

Reviewers' comments:

Reviewer's Responses to Questions

**Comments to the Author**

1. Is the manuscript technically sound, and do the data support the conclusions?

Reviewer #1: Yes

Reviewer #2: Yes

Reviewer #3: No

2. Has the statistical analysis been performed appropriately and rigorously? 

Reviewer #1: Yes

Reviewer #2: Yes

Reviewer #3: No

3. Have the authors made all data underlying the findings in their manuscript fully available?

Reviewer #1: Yes

Reviewer #2: Yes

Reviewer #3: Yes

4. Is the manuscript presented in an intelligible fashion and written in standard English?

Reviewer #1: Yes

Reviewer #2: Yes

Reviewer #3: No

5. Review Comments to the Author

Reviewer #1: 1- What about running time (execution time) of the method?

2- The "Related Work" section is missing. A new section should be added between "Introduction" and "Section 2".

Some parts of the "Introduction" section can be moved to the new section.

In addition, providing a table that summarizes the related work would increase the understandability of the difference from the previous studies in the related works section.

3- There are many symbols. A notation table can be added to give all symbols and their meanings.

4- The symbols in the text should be italic.

For example:

- "The features are as follows (with 0 ≤ X ≤ 4)"

- "the agent defines 5 features (N = 5)"

- ... etc.

Reviewer #2: The paper is generally well written and in my opinion above the threshold for publication, however, there are certain issues that should be addressed prior to it. Therefore I recommend the manuscript to be accepted pending minor revision.

Grammar/spelling/other:

1) Page 4 => "Section2" should be replaced by "Section 2", "making trading (3, 8) ." should be replaced by "making trading (3, 8)."

2) Please proofread the whole manuscript once again

3) Page 7=> "o" should be changed to "of"

4) Page 8 => "The [(, ) + max ′ (′, ′)] is referred to as the target Q-Value" => add the word "TERM" before "is referred"

5) Page 19 => "Error! Reference source not found.Table" => please amend this

6) "Linear in Inventory with Inventory Constraints (Linearly constant spread) (10): LIIC is also a

constant linear spread algorithm that places first-level quotes on both sides of the market. It differs

from FOIC in its inventory offset strategy to reduce risk: in LIIC, when an positive (negative)

inventory threshold is reached, the quantity of the buy (sell) orders is decreased linearly." => this is false, that is not how the LIIC works. LIIC does not place first-level quotes on both sides of the market but rather determines the offsets as a linear function of the current inventory. Please change this.

Literature review:

1) I recommend also referencing the following papers:

Guéant, O. (2017). Optimal market making. Applied Mathematical Finance, 24(2), 112-154.

- provides a thorough overview of market making

Gašperov, Bruno, et al. "Reinforcement Learning Approaches to Optimal Market Making." Mathematics 9.21 (2021): 2689.

- provides a survey of RL approaches to market making

Patel, Yagna. "Optimizing market making using multi-agent reinforcement learning." arXiv preprint arXiv:1812.10252 (2018).

- application of multi-agent RL to market making

Gašperov, Bruno, and Zvonko Kostanjčar. "Deep Reinforcement Learning for Market Making Under a Hawkes Process-Based Limit Order Book Model." IEEE Control Systems Letters (2022).

- consideration of more sophisticated limit order book and market making models

Comments:

1) Page 5 => explain that the Poisson rate at which the orders are executed at a Poisson rate decreases linearly with the spread

2) Page 6 => "The AS model generates bid and ask quotes that are optimal only if all of the assumptions it relies on

are accurate." => not really, this statement is too strong - the AS model ignores the existence of the bid-ask spread and is ill-suited for market making (MM) on stocks and better suited for MM on quote-driven markets (bonds etc.) Elaborate on this a bit more.

3) Page 10 => "last N ticks (orders entered in the order book)." Why do you only mention order arrivals here, and not cancelations as well, since they also change the state of the limit order book? Add cancelations as well.

Questions:

1) The authors rely on the use of a genetic algorithm to determine the parameters of the AS formulas. Why are the advantages of using this as compared to simply calibrating the parameters, for example in accordance with this procedure laid out here: https://www.theses.fr/2015PA066354.pdf (PhD thesis by Joaquin FERNANDEZ TAPIA). Please elaborate on it in the body of the manuscript.

2) Page 8 => "actions performed by our RL agent are the setting of the AS parameter values" => why AS and not Gueant-Lehalle-Fernandez-Tapia approximations which do not include the terminal time T and are considered SOTA and used by the banks?

3) Page 8 => "power to provide optimal quotes in the ideal case" => the AS procedure generates approximations to the optimal quotes and not the optimal quotes. Again, the statement is too strong and should be qualified.

4) Page 10 => "The agent is going to repeat the chosen action at every orderbook tick that occurs throughout the time step." Please explain this more thoroughly. Does this mean that, if the action as chosen at the first tick and the order gets executed instantly (and hence the inventory changes), the same action is repeated regardless of the fact that the inventory has changed? If so, why is this not problematic?

5) Why do you opt for discretized large action space instead of simply using a continuous action space and an appropriate RL algorithm, especially given there is a great selection of RL algorithms capable of tackling continuous action spaces? Please elaborate a bit.

6) Page 10 => "Greater importance was assigned to features that had a larger effect on the reward". How do you precisely determine the effect on the reward? What do you use as labels?

7) Page 10 => "The agent describes itself by the amount of inventory it holds and the reward it receives after

performing actions" => please elaborate on the importance of using reward as part of the state space as it seems unorthodox

8) Page 13 => "set {0.01, 0.1, 0.2, 0.9}". How do you obtain the values, are they chosen arbitrarily?

9) Page 16 => "There are two basic parameters to be determined in our direct Avellaneda-Stoikov model" => why only two, what about the parameter A?

Additional comments:

1) Are you acquainted with this GitHub repo? (https://github.com/im1235/ISAC) The authors also seem to be directly controling the risk aversion parameter gamma, i.e., using it as the action.

Reviewer #3: The authors consider the problem of market making, and propose a reinforcement learning (RL) based approach to estimate the optimal choices for the hyperparameters of the Avellaneda-Stoikov (AS) procedure. The proposed approach was implemented using deep Q-learning with two neural network architectures, and tested against a static choice for the parameters using a genetic algorithm and two simple benchmark models. The authors report that, based on the financial performance metrics of the algorithms, the RL approach yields improved results.

Generally, the paper is very hard to read. Some of it is due to the many formatting mistakes, but the organization is also flawed and makes it very hard for the reader to understand the proposed methods and results. Some other issues are listed below:

- The double DQN approach is relatively poorly explained, especially when taking into account the fact that this is one of the central parts of the proposed methodology.

- When considering data and features, the truncation of the ranges to the interval [-1,1] is left somewhat vague - is this a regular machine learning normalization procedure (done only on training data) or something else?

- The choices of N=5 ticks for private indicators and N=3 for market indicators seem very arbitrary.

- When using the random forest (RF) model for variable selection, it is unclear what the RF classifiers are actually classifying (which output variable and labels were used)?

- The main benchmark is a static selection of parameters for the AS procedure (the Gen-AS method) - however, the RL based methods are re-trained every 4 hours. A much more fair comparison would be not to re-train the RL methods, or to re-train the Gen-AS method, since in this case it is not only the RL approach that may provide improvement, but also the dynamic choice of AS parameters through time.

- A significant issue are the reported statistical results - the Kruskal-Wallis and Mann-Whitney tests are not paired tests (to the reviewer's best knowledge, they assume independent samples). However, the data points themselsves (the performance metrics over the test period days) are paired - i.e., the results for the same day are not independent, since they use the same trading data. If paired versions of these tests are used, the results may not hold.

Other minor issues include:

- the open P&L for the 5-second action time step is defined as Psi(t_i), but the Psi variable is not used in eq. 14 - are these the same quantities?

- Table 8 is somewhat unclear (specifically, the sign comparisons)

- reference formatting is not consistend, and some referenced papers appear listed twice in bibliography

- additional formatting error such as referencing Section 0, "equations X and X", and several "Error! Reference source not found", capitalization of "python" etc.

6. PLOS authors have the option to publish the peer review history of their article (what does this mean?). If published, this will include your full peer review and any attached files.

Reviewer #1: No

Reviewer #2: No

Reviewer #3: No

---

## [Author Response · Author response to Decision Letter 0]

15 Jul 2022

Here we couldn't attached all the response to reviewers document so please go to the document to read all the answers

Thank you and sorry for the inconvenience

---

## [Decision Letter · Decision Letter 1]

8 Aug 2022

PONE-D-22-10287R1A reinforcement learning approach to improve the performance of the Avellaneda-Stoikov market-making algorithmPLOS ONE

Dear Dr. Falces Marin,

Thank you for submitting your manuscript to PLOS ONE. After careful consideration, we feel that it has merit but does not fully meet PLOS ONE’s publication criteria as it currently stands.  Although 2 reviewers consider that the manuscript is suitable of publication in its current stand, one of the reviewers still show some concerns that need to be addressed before to deserve this manuscript for publication. These concerns are referred to the methodological part of the research and the writing style. However, I am sure that the author will be able to solve these issues. Therefore, we invite you to submit a revised version of the manuscript that addresses the points raised during the review process.

We look forward to receiving your revised manuscript.

Kind regards,

J E. Trinidad Segovia

Section Editor

PLOS ONE

Reviewers' comments:

Reviewer's Responses to Questions

**Comments to the Author**

1. If the authors have adequately addressed your comments raised in a previous round of review and you feel that this manuscript is now acceptable for publication, you may indicate that here to bypass the “Comments to the Author” section, enter your conflict of interest statement in the “Confidential to Editor” section, and submit your "Accept" recommendation.

Reviewer #1: All comments have been addressed

Reviewer #2: All comments have been addressed

Reviewer #3: (No Response)

2. Is the manuscript technically sound, and do the data support the conclusions?

Reviewer #1: Yes

Reviewer #2: (No Response)

Reviewer #3: (No Response)

3. Has the statistical analysis been performed appropriately and rigorously? 

Reviewer #1: Yes

Reviewer #2: (No Response)

Reviewer #3: (No Response)

4. Have the authors made all data underlying the findings in their manuscript fully available?

Reviewer #1: Yes

Reviewer #2: (No Response)

Reviewer #3: (No Response)

5. Is the manuscript presented in an intelligible fashion and written in standard English?

Reviewer #1: Yes

Reviewer #2: (No Response)

Reviewer #3: (No Response)

6. Review Comments to the Author

Reviewer #1: The authors revised the manuscript adequately according to the reviewers comments.

The manuscript is now more qualified and clear.

I have no further comments.

I suggest accepting it for publication in present form.

Reviewer #2: (No Response)

Reviewer #3: The revised version of the paper addressed the reviewer comments. The author responses to some methodological questions are noted and these points are mostly accounted for by the authors. Some remaining issues are listed below.

The question of the truncation of the interval of possible state feature values remains open, or there seems to be some misunderstanding between the authors and the reviewer. For instance, how are market prices (or actually differences to the mid-price) truncated to the interval [-1,1]? Are they scaled by some scaling parameter beforehand - and what data is this parameter estimated from (in ML literature this is usually done using only the training data)? If not, how much data is lost by only using the price differences with absolute values smaller than 1? Moreover, how are quantities truncated? Also, if the market candle features are "divided by the open mid-price for the candle", does this mean that all of those higher than the mid-price would be would be truncated to 1? The methodology might be more sound than this, but the text simply does not offer answers to these questions.

In general, the legibility of the paper is hardly improved, and the revisions in this regards were mostly superficial. The reviewer can point in the directions and give some examples but it is simply impossible to list all of the specific details, and it should be on the authors to check the manuscript in detail.

For instance, even after comments about reference formatting, some references have missing publications, years, issues, or even author names (there are many examples, and the authors should take a detailed look into all references and citations). Also, there seems to be a large number of arxiv or SSRN preprints listed for references which are actually published, either as working papers by some institutions or even in peer reviewed journals (and thus the arxiv or SSRN version is not a proper citation). Some of these will most likely be handled by the editorial team, but the extent of the errors is too large, evidently due to the revisions made by authors being mostly superficial.

Some additional minor issues:

- there is no need to capitalize terms such as reinforcement learning, profit and loss, Q-learning, Q-value, etc.

- what does X in the subscript of the features denote, if t is used as the time instance above? Generally, using X as a time index is unusual and might lead to confusion.

7. PLOS authors have the option to publish the peer review history of their article (what does this mean?). If published, this will include your full peer review and any attached files.

Reviewer #1: No

Reviewer #2: No

Reviewer #3: No

---

## [Author Response · Author response to Decision Letter 1]

20 Sep 2022

We are glad the reviewer deems the changes we have made to the manuscript have answered most of the concerns identified.

Attached there is a document with all reponses.

Thank you

---

## [Decision Letter · Decision Letter 2]

19 Oct 2022

A reinforcement learning approach to improve the performance of the Avellaneda-Stoikov market-making algorithm

PONE-D-22-10287R2

Dear Dr. Falces Marin,

We’re pleased to inform you that your manuscript has been judged scientifically suitable for publication and will be formally accepted for publication once it meets all outstanding technical requirements.

Kind regards,

J E. Trinidad Segovia

Section Editor

PLOS ONE

Additional Editor Comments (optional):

Reviewers' comments:

Reviewer's Responses to Questions

**Comments to the Author**

1. If the authors have adequately addressed your comments raised in a previous round of review and you feel that this manuscript is now acceptable for publication, you may indicate that here to bypass the “Comments to the Author” section, enter your conflict of interest statement in the “Confidential to Editor” section, and submit your "Accept" recommendation.

Reviewer #3: All comments have been addressed

2. Is the manuscript technically sound, and do the data support the conclusions?

Reviewer #3: (No Response)

3. Has the statistical analysis been performed appropriately and rigorously? 

Reviewer #3: (No Response)

4. Have the authors made all data underlying the findings in their manuscript fully available?

Reviewer #3: (No Response)

5. Is the manuscript presented in an intelligible fashion and written in standard English?

Reviewer #3: (No Response)

6. Review Comments to the Author

Reviewer #3: (No Response)

7. PLOS authors have the option to publish the peer review history of their article (what does this mean?). If published, this will include your full peer review and any attached files.

Reviewer #3: No

---

## [Editor Report · Acceptance letter]

28 Oct 2022

PONE-D-22-10287R2 

A reinforcement learning approach to improve the performance of the Avellaneda-Stoikov market-making algorithm 

Dear Dr. Falces Marin:

I'm pleased to inform you that your manuscript has been deemed suitable for publication in PLOS ONE. Congratulations! Your manuscript is now with our production department. 

Kind regards, 

on behalf of

Dr. J E. Trinidad Segovia 

Section Editor

PLOS ONE